# Distributed DPHelmet: Differentially Private Non-interactive Convex Blind Averaging

## Abstract

Differentially private massively distributed learning poses one key challenge when compared to differentially private centralized learning, where all data are aggregated at one party: minimizing communication overhead while achieving strong utility-privacy tradeoffs. The minimal amount of communication for distributed learning is non-interactive communication, i.e., each party only sends one message.

In this work, we propose two differentially private, non-interactive, distributed learning algorithms in a framework called Secure Distributed DP-Helmet. This framework is based on what we coin blind averaging: each party locally learns and noises a model and all parties then jointly compute the mean of their models via a secure summation protocol (e.g., secure multiparty computation). The learning algorithms we consider for blind averaging are empirical risk minimizers (ERM) like SVMs and Softmax-activated single-layer perception (Softmax-SLP). We show that blind averaging preserves privacy if the models are averaged via secure summation and the objective function is smooth, Lipschitz, and strongly convex. We show that the objective function of Softmax-SLP fulfills these criteria, which implies leave-one-out robustness and might be of independent interest.

On the practical side, we provide experimental evidence that blind averaging for SVMs and Softmax-SLP can have a strong utility-privacy tradeoff: we reach an accuracy of $86\%$ on CIFAR-10 for $\varepsilon = 0.36$ and $1,000$ users and of $44\%$ on CIFAR-100 for $\varepsilon = 1.18$ and $100$ users, both after a SimCLR-based pre-training. As an ablation, we study the resilience of our approach to a strongly non-IID setting. On the theoretical side, we show that in the limit blind averaging hinge-loss based SVMs convergences to the centralized learned SVM. Our approach is based on the representer theorem and can be seen as a blueprint for finding convergence for other ERM problems like Softmax-SLP.

## 1 Introduction

Privacy-preserving massively distributed learning poses one key challenge when compared to centralized learning: minimizing communication overhead, especially the number of communication rounds, while achieving strong utility-privacy tradeoffs. Jayaraman et al. (2018) achieved strong utility-privacy tradeoffs for empirical risk minimization (e.g., SVMs) in a number of communication rounds logarithmical in the training iterations. Yet, even few rounds can be expensive when scaling to hundreds, thousands, or millions of participants. Ideally, every party only sends a single message. For such optimal non-interactive communication, no prior work has achieved utility-privacy tradeoffs comparable to centralized learning, not even for well-understood tasks like SVMs learning.

One proposal for scalable distributed learning is differentially private federated learning (DP-FL). In DP-FL, each party protects its local data by only submitting local model updates protected via strong DP guarantees. A central server then aggregates all incoming updates (McMahan et al., 2017; Abadi et al., 2016). While DP-FL's communication rounds are independent of the number of parties, they are proportional to the number of training iterations $M$. Moreover, DP-FL achieves a significantly weaker utility-privacy tradeoffs than central learning, already for hundrets of parties.

Computation- and communication-heavy cryptographic methods can achieve the same utility-privacy tradeoffs as the centralized setting. However, these methods generally incur a large communication or computation overhead. Recent work has introduced efficient secure summation methods, such as

MPC for addition only (Bell et al., 2020). Secure summation methods can, e.g., be applied to DP-FL and lead to privacy-utility tradeoff that match its central learning counterpart DP-SGD (Abadi et al., 2016). Yet, DP-FL is interactive: the number of secure summation invocations is proportional to $M$.

In this work, we introduce a non-interactive distributed learning framework for two smooth convex ERMs, for SVM and a Softmax-activated single layer perception (Softmax-SLP), which is useful for last-layer fine-tuning during transfer learning: Secure Distributed DP-Helmet. We focus on what we coin blind averaging: each party runs an unsynchronized local learning algorithm, then locally noises the models, and then averages the models via scalable secure summation. Secure summation based on MPC can incur a few communication rounds, yet we invoke it only once. If computation servers are used for secure summation, then clients need only send one message (Bogetoft et al., 2009); hence, we consider blind averaging non-interactive from the client's perspective.

**Contribution.** Our contribution is fourfold:

**(1) Output sensitivity suffices for strong privacy results in blind averaging.** We show a sufficient condition for privacy for blind averaging, if at least a fraction $t$ (e.g., $50\,\%$) of users is honest: a bound on the effect of exchanging any one data point on the locally trained model. Given a bounded local output sensitivity, the output sensitivity after secure summation coincides with centralized learning: $\mathcal{O}((\sum_i |D^{(i)}|)^{-1})$ for the $n$ local datasets $(D^{(i)})_{i=1}^n$. To avoid expensive distributed noise generation protocols, each user locally adds Gaussian noise, which when summed up results in Gaussian noise.

**(2) Softmax-layer learning satisfies an output sensitivity bound.** For multiple classes, an SVM-approach trains one SVM for each class (e.g., via one-versus-rest), which neither scales in utility since the classes cannot be balanced nor in privacy since as many sequential compositions are needed as there are classes. A Softmax-SLP solves this multi-class challenge and is often used for single-layer fine-tuning. Wu et al. (2017) show that it suffices to prove that the loss function is smooth, Lipschitz, and strongly convex. We prove these properties for Softmax-SLP learning and obtain the first output sensitivity bounds for Softmax-SLP learning, which might be of independent interest as it implies leave-one-out robustness for Softmax-SLP learning.

**(3) Experiments illustrate strong utility-privacy tradeoffs.** We show that distributed SVMs and distributed Softmax-SLP training achieve competitive utility-privacy tradeoffs by evaluating them on CIFAR-10 and CIFAR-100. We utilize a feature extractor that has been pre-trained on ImageNet data (SimCLR by Chen et al. (2020b)). We observe an accuracy of $86\,\%$ on CIFAR-10 for $\varepsilon = 0.36$ and $1{,}000$ users and of $44\,\%$ on CIFAR-100 for $\varepsilon = 1.18$ and $100$ users. We also evaluate strongly non-IID scenarios where each party solely holds data from one class. As the non-interactivity and the feasibility of the local learning algorithms enable massive scalability but the benchmark datasets are limited, we extrapolate compelling utility-privacy results for millions of users.

**(4) Blind averaging for SVM learning: convergence in the limit & sufficient condition.** On the theoretical side, we derive a sufficient condition for strong utility from the representer theorem (Argyriou et al., 2009) which works for a large class of regularized ERM tasks. For that we utilize the dual representation of ERMs which leads to a characterization of the utility implications of any learning task that utilizes blind averaging. For SVMs, we use this characterization to prove graceful convergence in the limit to the best model for the combined local datasets $\mho$. For other ERMs, our sufficient condition for blind averaging leads to a precise formulation of an open problem that needs to be solved to prove convergence. We consider this precise formulation of the open problem an important step forward toward better understanding blind averaging beyond SVM learning.

## 2 RELATED WORK

Here we discuss work that is most related to our results; in Appx. D we discuss related work in more detail. In Tbl. 1 we detail our utility-bound improvement in comparison to prior work for SVM-based algorithms and inherently multi-class Softmax-SLP algorithms. Our work matches the utility-privacy tradeoff of centralized training with only one invocation of secure summation. We demonstrate the utility bounds theoretically for SVMs and indicate them empirically for Softmax-SLPs. With only one invocation, DP-Helmet is even realistic for Smartphone-based applications since we avoid issues stemming from multiple consecutive communication rounds like dropouts or unstable connectivity. When using Bell et al. (2020)'s construction for secure summation, the number of communication rounds is fixed to $4$ rounds, with each round's costs increased by only $\log^2(\text{n\_users})$. In comparison

Table 1: Comparison to related work for $n$ users with $m$ data points each: utility guarantee, DP noise scale, and number of MPC invocations. 'Utility: CC' denotes whether the local models converge to the centralized setting; for DPHelmet this reflects that blind averaging works. The convergence rate of Jayaraman et al. (2018, Output perturbation) depends on the dataset size whereas we truly converge with the number of iterations (cf. Thm. 14). All listed utility bounds are obtained if no noise is added. ($\checkmark$) denotes that this has been experimentally indicated, but no formally shown convergence.
∗: In DP-FL an untrusted aggregator combines the differentially private updates (users add noise and norm-clip those); it does not invoke MPC but needs a communication round per training iteration.

| SVM Algorithms | Utility: CC | Noise | MPC rounds |
| --- | --- | --- | --- |
| Jayaraman et al. (2018), gradient perturbation | $\checkmark$ | $\mathcal{O}(1/\mathbf{nm})$ | $\mathcal{O}(\log(nm))$ |
| Jayaraman et al. (2018), output perturbation | ($\checkmark$) | $\mathcal{O}(1/\mathbf{nm})$ | $\mathbf{1}$ |
| *Secure Distributed DP-Helmet: SVM (ours)* | $\checkmark$ | $\mathcal{O}(1/\mathbf{nm})$ | $\mathbf{1}$ |
| Softmax-SLPs Algorithms | | | |
| DP federated learning (DP-FL) | $\checkmark$ | $\mathcal{O}(1/m\sqrt{n})$ | ∗ |
| *Secure Distributed DP-Helmet: Softmax-SLP (ours)* | ($\checkmark$) | $\mathcal{O}(1/\mathbf{nm})$ | $\mathbf{1}$ |
| Baseline: Centralized training | $\checkmark$ | $\mathcal{O}(1/\mathbf{nm})$ | $\mathbf{0}$ |

---

**Algorithm 1:** SVM_SGD$(D, \xi, K)$ with hyperparameters $\xi := (h, c, \Lambda, R, M)$

**Data:** dataset $D := \left\{ (x_j, y_j) \right\}_{j=1}^N$ where $x_j$ is structured as $[1, x_{j,1}, \ldots, x_{j,p}]$;  set of classes $K$;
    Huber loss smoothness parameter $h \in \mathbb{R}_+$;  input clipping bound: $c \in \mathbb{R}_+$;  #iterations $M$;
    regularization parameter: $\Lambda \in \mathbb{R}_+$;  model clipping bound: $R \in \mathbb{R}_+$;

**Result:** models (1d intercepts with $p$-dimensional hyperplanes): $\left\{ f_M^{(k)} \right\}_{k \in K} \in \mathbb{R}^{(p+1) \times |K|}$

$clipped(x) := c \cdot x / \max(c, \|x\|)$;
$\mathcal{J}(f, D, k) := \frac{\Lambda}{2} f^T f + \frac{1}{N} \sum_{(x,y) \in D} \ell_{huber}\left(h, f^T clipped(x) \cdot y \cdot (1[y = k] - 1[y \neq k])\right)$;
**for** $k$ **in** $K$**:**
    **for** $m$ **in** $1, \ldots, M$**:**
        $f_m^{(k)} \leftarrow$ SGD$(\mathcal{J}(f_{m-1}^{(k)}, D, k), f_{m-1}^{(k)}, \alpha_m)$ on learning rate $\alpha_m = \min(\frac{1}{\beta}, \frac{1}{\Lambda m})$ and $\beta = \frac{1}{2h} + \Lambda$;
        $f_m^{(k)} := R \cdot f_m^{(k)} / \|f_m^{(k)}\|$;                  // projected SGD

---

to DP-FL, we have a 500-fold decrease in total communication cost for CIFAR-10 data: DP-FL has 1,920 rounds (40 epochs á 48 batches) of communication cost $\ell$, where $\ell$ is the model size (roughly 60,000 floats), while we have 4 rounds of cost $\log^2(n\_users) + \ell$ for roughly the same model size.

## 3 PRELIMINARIES

### 3.1 DIFFERENTIAL PRIVACY AND DP_SVM_SGD

Preliminaries of the Secure Summation protocol (Bell et al., 2020) as well as pre-training as a tactic to boost DP performance are available in Appx. C.2 and Appx. C.3 respectively. Intuitively, differential privacy (DP) (Dwork et al., 2006b) quantifies the protection of any individual's data within a dataset against an arbitrarily strong attacker observing the output of a computation on said dataset. Strong protection is achieved by bounding the influence of each individual's data on the resulting SVMs. For the (standard) definition of differential privacy we utilize in our proofs, we refer to Appx. C.1.

We consider Support Vector Machines (SVMs), which can be made strongly convex, thus display a unique local minimum and a lower bound on the growth of the optimization function. A differentially private SVM definition (DP_SVM_SGD) can be derived directly from the work of Wu et al. (2017) on empirical risk minimization using SGD-based optimization. They rely on a smoothed version of the hinge-loss: the Huber loss $\ell_{huber}$ (cf. Appx. C.5 for details). We additionally apply norm-clipping to all inputs. We use the one-vs-rest (OVR) method to achieve a multiclass classifier. Alg. 1 provides pseudocode for the sensitivity-bounded algorithm before adding noise.

In contrast to Wu et al. (2017), which assumes for each data point $\|x\| \le 1$, we use a generalization that holds for larger norm bounds $c > 1$: we assume $\|x\| \le c$, where $c$ is a hyperparameter of the learning algorithm SVM_SGD. As a result, the optimization function $\mathcal{J}$ is $c + R\Lambda$ Lipschitz (instead of $1 + R\Lambda$ Lipschitz as in Wu et al. (2017)) and $((c^2/2h + \Lambda)^2 + p\Lambda^2)^{1/2}$ smooth (instead of $1/2h + \Lambda$ smooth). Wu et al. (2017) showed a *sensitivity bound* for SVM_SGD from which we can conclude DP guarantees. The sensitivity proof follows from Wu et al. (2017, Lemma 8) with the Lipschitz constant $L = c + R\Lambda$, a smoothness $\beta = ((c^2/2h + \Lambda)^2 + p\Lambda^2)^{1/2}$ and a $\Lambda$-strong convexity.

Similarly, our work applies to $L_2$-regularized logistic regression where we adapt Alg. 1 with the optimization function $\mathcal{J}'(f, D) := \frac{\Lambda}{2} f^T f + \frac{1}{N} \sum_{(x,y) \in D} \ln(1 + \exp(-f^T \, clipped(x) \cdot y))$ which is $\Lambda$-strongly convex, $L = c + R\Lambda$ Lipschitz, and $\beta = ((c^2/4 + \Lambda)^2 + p\Lambda^2)^{1/2}$ smooth. We adapt the learning rate to accommodate the change in $\beta$ but have the same sensitivity as the classification case.

**Definition 1** (Sensitivity). *Let $f$ be a function that maps datasets to the p-dimensional vector space $\mathbb{R}^p$. The* sensitivity *of $f$ is defined as $\max_{D \sim_1 D'} \|f(D) - f(D')\|$, where $D \sim_1 D'$ denotes that the datasets $D$ and $D'$ differ in at most one element. We say that $f$ is an $s$-sensitivity-bounded function.*

**Lemma 2.** *With input clipping bound $c$, model clipping bound $R$, strong convexity factor $\Lambda$, and $N$ data points, the learning algorithm SVM_SGD of Alg. 1 has a sensitivity bound of $s = \frac{2(c+R\Lambda)}{N\Lambda}$ for each of the $|K|$ output models. This directly follows from Wu et al. (2017, Lemma 8).*

For sensitivity-bounded functions, there is a generic additive mechanism that adds Gaussian noise to the results of the function and achieves differential privacy, if the noise is calibrated to the sensitivity.

**Lemma 3** (Gaussian mechanism is DP (Theorem A.1 & Theorem B.1 in Dwork & Roth (2014))). *Let $q_k$ be functions with sensitivity $s$ on the set of datasets $\mathcal{D}$. For $\varepsilon \in (0,1)$, $c^2 > 2 \ln 1.25/(\delta/|K|)$, the Gaussian Mechanism $D \mapsto \{q_k(D)\}_{k \in K} + \mathcal{N}(0, (\sigma \cdot I_{(p+1) \times |K|})^2)$ with $\sigma \ge \frac{c \cdot s \cdot |K|}{\varepsilon}$ is $(\varepsilon, \delta)$-DP, where $I_d$ is the $d$-dimensional identity matrix.*

**Corollary 4** (Gaussian mechanism on SVM_SGD is DP). *With the $s$-sensitivity-bounded learning algorithm SVM_SGD (cf. Lem. 2), the dimension of each data point $p$, the set of classes $K$, and $\varepsilon \in (0,1)$, $DP\_SVM\_SGD(D, \xi, K, \sigma) := SVM\_SGD(D, \xi, K) + \mathcal{N}(0, (\sigma \cdot s \cdot I_{(p+1) \times |K|})^2)$ is $(\varepsilon, \delta)$-DP, where $\varepsilon \ge \sqrt{2 \ln 1.25/(\delta/|K|)} \cdot |K| \cdot 1/\sigma$ and $I_d$ is the $d$-dimensional identity matrix.*

There are tighter composition results (Meiser & Mohammadi, 2018; Sommer et al., 2019; Balle et al., 2020a) where $\varepsilon \in \mathcal{O}(\sqrt{|K|})$ which we do not formalize for brevity but follow in our experiments.

**Definition 5** (Configuration $\zeta$). *A configuration $\zeta(\mathcal{U}, t, T, s, \xi, \mho, i, N, K, \sigma)$ consists of a set of users $\mathcal{U}$ of which $t \cdot \mathcal{U}$ are honest, an $s$-sensitivity-bounded learning algorithm $T$ on inputs $(D, \xi, K)$, hyperparameters $\xi$, a local datasets $D^{(i)}$ of user $U^{(i)} \in \mathcal{U}$ with $N = \min_{i \in \{1, \ldots, |\mathcal{U}|\}} |D^{(i)}|$ and $\mho = \bigcup_i^{|\mathcal{U}|} D^{(i)}$, a set of classes $K$, and a noise multiplier $\sigma$. $avg(T)$ is the aggregation of $|\mathcal{U}|$ local models of algorithm $T$: $avg_i(T(D^{(i)})) = \frac{1}{|\mathcal{U}|} \sum_{i=1}^{|\mathcal{U}|} T(D^{(i)}, \xi, K)$. If unique, we simply write $\zeta$.*

## 4 Secure Distributed DP-Helmet: System Design

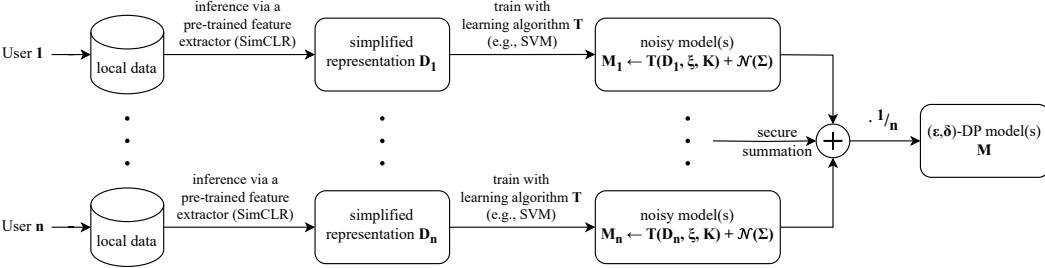

Figure 1: Schematic overview of Secure Distributed DP-Helmet. Each user locally applies a pre-trained feature extractor (SimCLR) to their data, then trains a model, e.g. an SVM, via a learning algorithm $T$, which is noised with a spherical $\Sigma$-parameterized Gaussian. A single secure summation step results in an averaged and $(\varepsilon, \delta)$-DP model. $\xi$ denotes hyperparameters and $K$ a set of classes.

---

**Algorithm 2:** Secure Distributed DP-Helmet. For $T$ = SVM_SGD (cf. Alg. 1), we have $s = \frac{2(c+R\Lambda)}{N\Lambda}$ and hyperparameters $\xi := (h, c, \Lambda, R, M)$. $\pi_{SecSum}$ as in Def. 18.

---

**def** `Client Secure Distributed DP-Helmet` ($D$, $|\mathcal{U}|$, $K$, $T$, $t$, $\xi$, $\sigma$)**:**

> **Data:** local dataset $D$ with $N = |D|$; #users $|\mathcal{U}|$; set of classes K; learning algorithm $T$;
> ratio $t$ of honest users; hyperparameters $\xi$; noise multiplier $\sigma$
>
> **Result:** DP-models (intercepts with $p$-dimensional hyperplanes): $M_{priv} := \left\{ f_{priv}^{(k)} \right\}_{k \in K}$
>
> $M_{odel} \leftarrow T(D, \xi)$;                         // $T$ is $s$-sensitivity-bounded
> $M_{priv} \leftarrow M_{odel} + \mathcal{N}(0, (\tilde{\sigma} \cdot s \cdot I_{p+1 \times |K|})^2)$ **with** $\tilde{\sigma} := \sigma \cdot {}^1/\sqrt{t \cdot |\mathcal{U}|}$;
> Run the client code of a secure summation protocol $\pi_{SecSum}$ on input ${}^{M_{priv}}/|\mathcal{U}|$;

**def** `Server Secure Distributed DP-Helmet` ($\mathcal{U}$)**:**

> **Data:** users $\mathcal{U}$
> **Result:** empty string
> Run the server protocol of $\pi_{SecSum}$;

---

We here present the system design of Secure Distributed DP-Helmet in detail (cf. Alg. 2) including its privacy properties. A schematic overview is illustrated in Fig. 1. Each user holds a small dataset while all users jointly learn a model. There are two scenarios: first, each person contributes one data point to a user who is a local aggregator, e.g., a hospital (differential privacy, see Fig. 3); second, each user is a person and contributes a small dataset (local DP, see Fig. 5 for $\Upsilon = 50$).

Consider a set of users $\mathcal{U}$, each with a local dataset $D$ of size $N = |D|$ that already is in a sufficiently simplified representation by the SimCLR pre-taining feature extractor (Chen et al., 2020a;b). The users collectively train a $(\varepsilon, \delta)$-DP model using a learning algorithm $T$ that is $s$-sensitivity-bounded as in Def. 1. An example for $T$ is SVM_SGD (cf. Alg. 1), which has $s = \frac{2(c+R\Lambda)}{N\Lambda}$.

Alg. 2 follows the scheme of Jayaraman et al. (2018): First, each user separately trains a non-private model $M_{odel}$, using $T$ and the hyperparameters $\xi$, e.g., $\xi := (h, c, \Lambda, R, M)$ for SVM_SGD (Wu et al., 2017). Next, each user adds to $M_{odel}$ Gaussian noise scaled with $s$ and ${}^1/\sqrt{t \cdot |\mathcal{U}|}$, where $t \cdot |\mathcal{U}|$ is the number of honest users in the system. Together the users then run a secure summation protocol $\pi_{SecSum}$ as in Def. 18 where the input of each user is the noised model, which is scaled down by the number of users to yield the average model. Thanks to secure summation we show centralized-DP guarantees with noise in the order of $\mathcal{O}(|\mathcal{U}|^{-1} N^{-1})$ within a threat model akin to that of federated learning with differential privacy. For privacy accounting, we use tight composition bounds like Meiser & Mohammadi (2018); Sommer et al. (2019); Balle et al. (2020a).

**Threat model & security goals.**    For our work, we assume passive, collaborating attackers that follow our protocol. We assume that a fraction of at least $t$ users are honest (say $t = 50\%$). The adversary is assumed to have full knowledge about each user's dataset, except for one data point of one user. Our privacy goals are $(\varepsilon, \delta)$-differential privacy (protecting single samples) and $(\varepsilon, \delta)$-$\Upsilon$-*group differential privacy* (protecting all samples of a user at once) respectively, depending on whether each user is a local aggregator or a person. Note that even passive adversaries can collude and exchange information about the randomness they used in their local computation. To compensate for untrustworthy users, we adjust the noise added by each user according to the fraction of honest users $t$; e.g., if $t = 50\%$, then we double the noise to satisfy our guarantees.

## 4.1 Security of Secure Distributed DP-Helmet

First, we derive a tight output sensitivity bound. A naïve approach would be to release each individual predictor, determine the noise scale proportionally to $\tilde{\sigma} := \sigma$ (cf. Cor. 12), showing $(\varepsilon, \delta)$-DP for every user. We can save a factor of $|\mathcal{U}|^{1/2}$ by leveraging that $|\mathcal{U}|$ is known to the adversary and we have at least $t = 50\%$. Consequently, local noise of scale $\tilde{\sigma} := \sigma \cdot {}^1/\sqrt{t \cdot |\mathcal{U}|}$ is sufficient for $(\varepsilon, \delta)$-DP.

**Lemma 6** (Privacy amplification via averaging). *For a configuration $\zeta$, Secure Distributed DP-Helmet($\zeta$) of Alg. 2 without noise, $avg_i(T(D^{(i)}))$, has a sensitivity of $s \cdot {}^1/|\mathcal{U}|$ for each class $k \in K$.*

The proof is placed in Appx. H. Having bounded the sensitivity of the aggregate to $s \cdot {}^1/|\mathcal{U}|$, we show that locally adding noise per user proportional to $\sigma \cdot s \cdot {}^1/\sqrt{|\mathcal{U}|}$ and taking the mean is equivalent to only centrally adding noise proportional to $\sigma \cdot s \cdot {}^1/|\mathcal{U}|$ (as if the central aggregator was honest).

---

**Algorithm 3:** Softmax_SLP_SGD$(D, \xi, K)$ with hyperparameters $\xi := (c, \Lambda, R, M)$

---

**Data:** dataset $D := \{(x_j, y_j)\}_{j=1}^{N}$ where $x_j$ is structured as $[1, x_{j,1}, \ldots, x_{j,p}]$;   set of classes $K$;
  #iterations $M$;
  input clipping bound: $c \in \mathbb{R}_+$; regularization parameter: $\Lambda \in \mathbb{R}_+$; model clipping bound: $R \in \mathbb{R}_+$;

**Result:** a model with intercept: $f_M \in \mathbb{R}^{(p+1) \times |K|}$

$clipped(x) := c \cdot x / \max(c, \|x\|)$;

$\mathcal{J}_{\text{softmax}}(f, D) := \frac{\Lambda}{2} \sum_{k=1}^{K} (f^T f)_k + \frac{1}{N} \sum_{(clipped(x), y) \in D} - \sum_{k=1}^{K} y_k \log \frac{\exp((f^T x)_k)}{\sum_{j=1}^{K} \exp((f^T x)_j)}$;

**for** $k$ **in** $K$:
  **for** $m$ **in** $1, \ldots, M$:
    $f_m^{(k)} \leftarrow SGD(\mathcal{J}_{\text{softmax}}(f_m, D, k), \alpha_m)$, with learning rate $\alpha_m := \min(\frac{1}{\beta}, \frac{1}{\Lambda m})$ and
    $\beta = \sqrt{(d+1)K\Lambda^2 + 0.5(\Lambda + c^2)^2}$;
    $f_m^{(k)} := R \cdot f_m^{(k)} / \|f_m^{(k)}\|$;                     // projected SGD

---

**Lemma 7.** *For configuration $\zeta$ and noise scale $\tilde{\sigma}$: $\frac{1}{|\mathcal{U}|} \sum_{i=1}^{|\mathcal{U}|} \mathcal{N}(0, (\tilde{\sigma} \cdot 1/\sqrt{|\mathcal{U}|})^2) = \mathcal{N}(0, (\tilde{\sigma} \cdot 1/|\mathcal{U}|)^2)$.*

The proof is placed in Appx. I. We can now prove differential privacy for Secure Distributed DP-Helmet of Alg. 2 where we have noise scale $\tilde{\sigma} := \sigma \cdot 1/\sqrt{t \cdot |\mathcal{U}|}$ and thus $\varepsilon \in \mathcal{O}(s/\sqrt{t \cdot |\mathcal{U}|})$.

**Theorem 8** (Main Theorem, simplified). *For a configuration $\zeta$ as in Def. 5, Secure Distributed DP-Helmet$(\zeta)$ of Alg. 2 satisfies computational $(\varepsilon, \delta + \nu)$-DP with $\varepsilon \geq \sqrt{2 \ln 1.25/(\delta/|K|)} \cdot |K| \cdot 1/\sigma$ and a function $\nu$ negligible in the security parameter used in $\pi_{SecSum}$.*

The full statement and proof are in Appx. J. Simplified, the proof follows by the application of the sensitivity (cf. Lem. 6) to the Gauss Mechanism (cf. Lem. 3) where the noise is applied per user (cf. Lem. 7). If each user contributes 50 data points and we have 1000 users, $N \cdot |\mathcal{U}| = 50{,}000$.

Next, we show how to protect the entire dataset of a single user (e.g., for distributed training via smartphones). The sensitivity-based bound on the Gaussian mechanism (see Appx. M) directly implies that we can achieve strong $\Upsilon$-group privacy results, which is equivalent to local DP.

**Corollary 9** (Group-private variant). *For a configuration $\zeta$ as in Def. 5, Secure Distributed DP-Helmet$(\zeta)$ of Alg. 2 satisfies computational $(\varepsilon, \delta + \nu)$, $\Upsilon$-group DP with $\varepsilon \geq \Upsilon \cdot \sqrt{2 \ln 1.25/(\delta/|K|)} \cdot |K| \cdot 1/\sigma$ for $\nu$ as above: for any pair of datasets $\mho, \mho'$ that differ at most $\Upsilon$ many data points,*

$$Secure\ Distributed\ DP\text{-}Helmet(\zeta(\ldots, \mho, \ldots)) \approx_{\varepsilon, \delta} Secure\ Distributed\ DP\text{-}Helmet(\zeta(\ldots, \mho', \ldots))$$

Cor. 9 generalizes to a more comprehensive Cor. 10 that is data oblivious. If the norm of each model is bounded, then Secure Distributed DP-Helmet can apply on the granularity of users instead of that of data points. This method enables the use of other SVM optimizers or Logistic Regression and can render a tighter sensitivity bound than SVM_SGD for certain settings of $\Upsilon$ or data points per user $N$. In particular, the training procedure of each base learner does not need to satisfy differential privacy.

**Corollary 10.** *Given a learning algorithm $T$, we say that $T$ is $R$-norm bounded if for any input dataset $D$ with $N = |D|$, any hyperparameter $\xi$, and all classes $k \in K$, $\|T(D, \xi, k)\| \leq R$. Any $R$-norm bounded learning algorithm $T$ has a sensitivity $s = 2R$. In particular, $T + \mathcal{N}(0, (\sigma \cdot s \cdot I_d)^2)$ satisfies $(\varepsilon, \delta)$, $\Upsilon$-group differential privacy with $\Upsilon = N$ and $\varepsilon \geq \sqrt{2 \ln 1.25/(\delta/|K|)} \cdot |K| \cdot 1/\sigma$, where $\mathcal{N}(0, (\sigma \cdot s \cdot I_d)^2)$ is spherical multivariate Gaussian noise and $\sigma$ a noise multiplier.*

The proof is in Appx. K. Here the number of local data points $N$ can vary among the users.

**Softmax_SLP_SGD.**    We also show differential privacy for a softmax-activated single-layer perception in Alg. 3 by showing that its objective function is $\Lambda$-strongly convex (cf. Thm. 26), $L = \Lambda R + \sqrt{2}c$ Lipschitz (cf. Thm. 27), and $\beta = \sqrt{(d+1)|K|\Lambda^2 + 0.5(\Lambda + c^2)^2}$ smooth (cf. Thm. 28). Then, the sensitivity directly follows from Wu et al. (2017, Lemma 8).

**Theorem 11** (Softmax sensitivity). *Given a configuration $\zeta$ as in Def. 5, the learning algorithm Softmax_SLP_SGD of Alg. 3 has a sensitivity bound of $s = \frac{2(\Lambda R + \sqrt{2}c)}{N\Lambda}$ for the output model.*

**Corollary 12** (Gauss on Softmax_SLP_SGD is DP). *Given a configuration $\zeta$ as in Def. 5, with the $s$-sensitivity-bounded learning algorithm Softmax_SLP_SGD (cf. Thm. 11), the dimension of each*

*data point p, and* $\varepsilon \in (0,1)$, *DP_Softmax_SLP_SGD*$(D, \xi, K, \sigma) := $ *Softmax_SLP_SGD*$(D, \xi, K) + \mathcal{N}(0, (\sigma \cdot s \cdot I_{(p+1) \times |K|})^2)$ *is* $(\varepsilon, \delta)$-*DP, where* $\varepsilon \geq \sqrt{2 \ln 1.25/\delta} \cdot 1/\sigma$ *and* $I_d$ *is the identity matrix.*

## 5 SECURE DISTRIBUTED DP-HELMET: NON-INTERACTIVE BLIND AVERAGE

The core idea of Secure Distributed DP-Helmet is to locally train models and compute a blind average without further synchronizing or fine-tuning the models: $\text{avg}(T)$. To show that such a non-interactive training is useful, we provide a utility bound on the blind averaging procedure. For that bound, we (1) reduce the utility requirement of blindly averaging a regularized empirical risk minimizer (ERM) $T$ to the coefficients $\alpha$ of the dual problem of $T$, (2) leverage the dual problem to show that for a hinge-loss linear SVM trained with SGD $T = $ HINGE_SVM_SGD $\text{avg}(T)$ gracefully convergences in the limit to the best model for the combined local datasets $\mho$: $\mathbb{E}\left[\mathcal{J}(\text{avg}_i(\text{HINGE\_SVM}(D^{(i)})), \mho, \_) - \inf_f \mathcal{J}(f, \mho, \_)\right] \in \mathcal{O}(1/M)$ for $M$ many local training rounds and objective function $\mathcal{J}$. Thus, convergence also holds in a strongly non-iid scenario which we illustrate with an example in Fig. 2 where each user only has access to one class.

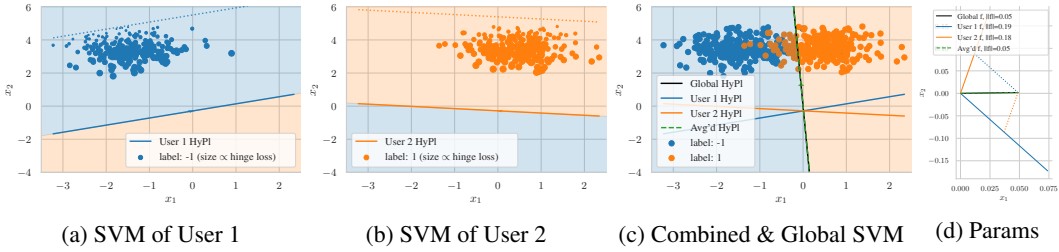

(a) SVM of User 1     (b) SVM of User 2     (c) Combined & Global SVM     (d) Params

Figure 2: Strongly biased local datasets: Local SVMs hyperplanes (solid) and their margins (dotted) on a point cloud $x$ for user 1 (a) and user 2 (b). After averaging, they approximate the global SVM trained on the combined point cloud (c). The Normal vectors $f$ of each respective SVM illustrate the average (d). Hyperparameters: $\Lambda = 20, R = 1, c = 5, N = 500, \text{bs} = 25, \text{epochs} = 500$.

We now show in Lem. 13 in Appx. L.1 that if each converged hinge-loss linear SVM $T(D^{(i)}, \xi, k) = \text{argmin}_f \frac{1}{N} \sum_{(x,y) \in D^{(i)}} \max(0, 1 - yf^T x) + \Lambda \|f\|^2$ on a local dataset $D^{(i)}$ has support vectors $V^{(i)}$ then the average of all these locally trained SVMs $\text{avg}_i(T(D^{(i)}))$ has support vectors $V = \bigcup_{i=1}^{|\mathcal{U}|} V^{(i)}$.

**Lemma 13** (Support Vectors of averaged SVM). *Given a configuration* $\zeta$ *as in Def. 5, a locally trained model of learning algorithm* $T$, *where* $T(D^{(i)}, \xi, k) = \text{argmin}_f \frac{1}{N} \sum_{(x,y) \in D^{(i)}} \max(0, 1 - f^T xy \cdot (1[y = k] - 1[y \neq k])) + \Lambda \|f\|^2$ *comprises a hinge-loss linear SVM, has the support vectors* $V^{(i)} = \left\{ (x, y) \in D^{(i)} \mid T(D^{(i)})^T xy \leq \|T(D^{(i)})\|^{-1} \right\}$. *Then, the average of these locally trained models* $\text{avg}_i(T(D^{(i)}))$ *has the support vectors* $V = \bigcup_{i=1}^{|\mathcal{U}|} V^{(i)}$.

If an SVM of the combined local datasets $\mho$ has the same support vectors as the average of local SVMs then both models converge due to the hinge loss definition (cf. Thm. 14 in Appx. L.2). Such a scenario occurs e.g. if the regularization is high and thus the margin is large enough such that all data points are within the margin and thus support vectors.

**Theorem 14** (Averaging locally trained SVM converges to a global SVM). *Given a configuration* $\zeta$ *as in Def. 5, there exists a regularization parameter* $\Lambda$ *such that the average of locally trained models* $\text{avg}_i(T(D^{(i)}))$ *with a hinge-loss linear SVM as objective function* $\mathcal{J}$ *trained with projected subgradient descent using weighted averaging (PGDWA),* $T = $ *HINGE_SVM_PGDWA, converges with the number of local iterations* $M$ *to the best model for the combined local datasets* $\mho$, *i.e.*

$$\mathbb{E}\left[\mathcal{J}(\text{avg}_i(\textit{HINGE\_SVM\_PGDWA}(D^{(i)})), \mho, \_) - \inf_f \mathcal{J}(f, \mho, \_)\right] \in \mathcal{O}(1/M).$$

The reason the average of SVMs has the union of the local support vectors as support vectors is that the average of SVMs has the union of the local dual coefficients as dual coefficients $\alpha$ (cf. Cor. 23). This corollary holds not only for hinge-loss linear SVMs but for a broad range of regularized empirical risk minimizes (ERM) for which the representer theorem holds (cf. Thm. 20 Argyriou et al. (2009)), including a converged SVM_SGD and Softmax_SLP_SGD. For limitations, we refer to Sec. 7.

## 6 EXPERIMENTAL RESULTS

We analyze three experimental research questions: (RQ1) How does Secure Distributed DP-Helmet compare to the strongest alternative, DP-SGD-based federated learning? We examine two dimensions for our comparisons: (RQ1.1) If each user has a set number of data points, how does performance compare when the number of users increases (cf. Fig. 4 and Fig. 3, middle)? (RQ1.2) If we keep the number over overall data points the same, how does distributing them impact performance (cf. Fig. 3, top and bottom)? (RQ2) How robust is our performance if the local datasets of users differ significantly, e.g. they are strongly biased non-iid (cf. Tbl. 2)? (RQ3) How do the learning algorithms DP_SVM_SGD and DP_Softmax_SLP_SGD perform in a centralized setting (cf. Appx. G)?

**Pretraining.** We used a SimCLR pretrained model[1] on ImageNet ILSVRC-2012 (Russakovsky et al., 2015) for all experiments (cf. Fig. 7 in the appendix for an embedding view). It is built with a ResNet152 with selective kernels (Li et al., 2019) architecture including a width multiplier of 3 and it has been trained in the fine-tuned variation of SimCLR where $100\%$ of ImageNet's label information has been integrated during training. Overall, it totals $795\,M$ parameters and achieves $83.1\%$ classification accuracy (1000 classes) when applied to a linear prediction head. In comparison, a supervised-only model of the same size would only achieve $80.5\%$ classification accuracy.

**Sensitive Dataset.** CIFAR-10 and CIFAR-100 (Krizhevsky, 2009) act as our sensitive datasets, as they are frequently used as benchmark datases in differential privacy literature. Both consist of 60,000 thumbnail-sized, colored images of 10 or 100 classes.

**Evaluation.** The model performance is delineated fourfold: First, we evaluated a benchmark

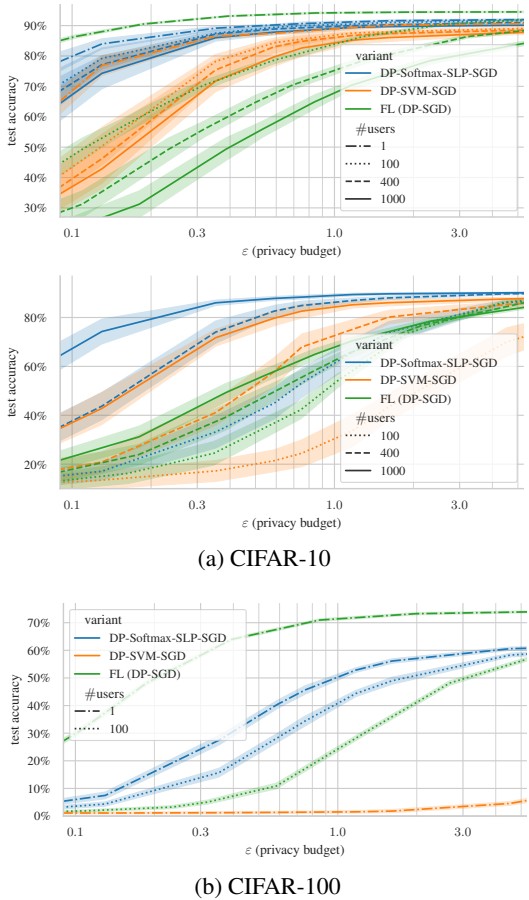

(a) CIFAR-10

(b) CIFAR-100

Figure 3: Classification accuracy compared to privacy budget $\varepsilon$ (in $\log_{10}$-scale) of Secure Distributed DP-Helmet (cf. Sec. 4) and DP-SGD-based federated learning (FL) ($\delta = 10^{-5}$). (top, bottom) We use all available data points of the dataset for each line, spreading them among a differing number of users. (middle) Different numbers of users with 50 data points per user.

scenario in Fig. 3 (top and bottom) to compare our Secure Distributed DP-Helmet (cf. Sec. 4) in the DP_SVM_SGD as well as DP_Softmax_SLP_SGD variant to a DP-SGD-based federated learning approach (DP-FL) on a single layer perceptron with softmax loss. There the approximately same number of data points is split across a various number of users ranging from 1 to 1000. Second, we also evaluated a realistic scenario in Fig. 3 (middle) where we fixed the number of data points per user and report the performance increase obtained with more partaking users. Fig. 4 depicts the setting of Fig. 3 (middle) for a fixed privacy budget. Third, we evaluated a strongly biased non-iid scenario where each user has exclusive access to one class (cf. Tbl. 2).

The experiments lead to four conclusions: (RQ1.1) First, performance improves with an increasing number of users (cf. Fig. 3 (middle)). Although DP_Softmax_SLP_SGD training performs subpar to DP-FL for few users, it takes off after about 100 users on both datasets due to its vigorous performance gain with the number of users (cf. Fig. 4). Our scalability advantage with the number of users becomes especially evident when considering significantly more users (cf. Fig. 5 in Appx. A).

[1]accessible at `https://github.com/google-research/simclr`, Apache-2.0 license

Table 2: Secure Distributed DP-Helmet– strongly biased non-IID experiments for $\varepsilon = 1.172$: each user has exclusive access to only one class. We compare the accuracy and difference in percentage points (pp) to the performance in our regular experiments. Similar to Fig. 5 in Appx. A, we extrapolate the accuracy on datasets 67 times larger by using a lower noise magnitude.

| | | ACCuracy on dataset | |
| Blindly averaging variants | dataset multiplier | CIFAR-10 | CIFAR-100 |
|---|---|---|---|
| DP_SVM_SGD | 1x | 85 % ACC ($-2$ pp) | – |
| DP_SVM_SGD | 67x | 87 % ACC ($-3$ pp) | – |
| DP_Softmax_SLP_SGD | 1x | 42 % ACC ($-49$ pp) | 1.6 % ACC ($-43$ pp) |
| DP_Softmax_SLP_SGD | 67x | 88 % ACC ($-4$ pp) | 58 % ACC ($-4$ pp) |

Here, DP-guarantees of $\varepsilon \leq 5 \cdot 10^{-5}$ become plausible with at least $87\%$ prediction performance for a task like CIFAR-10. Alternatively, leveraging Cor. 9 we can consider a local DP scenario (with $\Upsilon = 50$) without a trusted aggregator, yielding an accuracy of $87\%$ for $\varepsilon = 1 \cdot 10^{-4}$. (RQ1.2) Second, if we globally fix the number of data points (cf. Fig. 3 (top and bottom)) that are distributed over the users, Secure Distributed DP-Helmet's performance degrades more gracefully than that of DP-FL. Thm. 14 supports the more graceful decline; it states that averaging multiple of an SVM similar to SVM_SGD converges for a large enough regularizer to the optimal SVM on all training data. In absolute terms, the accuracy is better for a smaller regularizer which is visible by the remaining discrepancy in Fig. 3 (top and bottom) between the number of users. Blind averaging also leads to a graceful decline for DP_Softmax_SLP_SGD. The difference between 1 and 100 users is largely due to our assumption of $t = 50\%$ dishonest users, which means noise is scaled by a factor of $\sqrt{2}$. In comparison, DP-FL performs worse the more users $\mathcal{U}$ partake as the noise scales with $\mathcal{O}(|\mathcal{U}|^{1/2})$.

(RQ2) In a strongly biased non-iid scenario like Tbl. 2, we observe that on CIFAR-10 the utility decline of DP_SVM_SGD is still small whereas DP_Softmax_SLP_SGD needs more users for a similar utility preservation since it is more sensitive to noise.

(RQ3) We refer to Appx. G for an ablation study in the centrally trained setting for our learning algorithms DP_SVM_SGD and DP_Softmax_SLP_SGD and other DP learners like DP-SGD (Abadi et al., 2016). In this setting, DP_SVM_SGD performs worse than DP_Softmax_SLP_SGD while DP-SGD outperforms both: for $\varepsilon = 0.59$ on CIFAR-10, DP_SVM_SGD has an accuracy of $87.4\%$ and DP_Softmax_SLP_SGD has $90.2\%$ while DP-SGD has $93.6\%$. For $\varepsilon = 1.18$ on CIFAR-100, DP_SVM_SGD has an accuracy of $1.5\%$ and DP_Softmax_SLP_SGD has an accuracy of $52.8\%$ while DP-SGD has $71.6\%$. We reckon that this difference is mostly due to DP_Softmax_SLP_SGD's multi-class approach with one sensitivity bound for all classes and DP-SGD's joint learning of all classes as well as noise-correcting property from its iterative noise application.

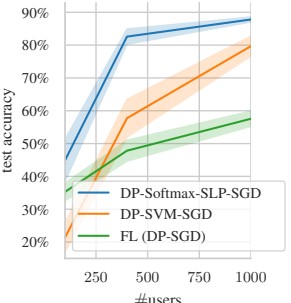

Figure 4: CIFAR-10 classification accuracy vs. #*users* with 50 data points per user for $\varepsilon = 0.5885$, $\delta = 10^{-5}$. Values for FL are interpolated.

**Computation costs.** For Secure Distributed DP-Helmet with $1{,}000$ users and a model size $l \approx 100{,}000$ for CIFAR-10, we need less than $0.2\,s$ for the client and $40\,s$ for the server, determined by extrapolating the experiments of Bell et al. (2020, Table 2).

**Experimental setup.** Appx. E describes our experimental setup.

## 7 LIMITATIONS & DISCUSSION

**Limitations of blind averaging** For unfavorable datasets, blind averaging leads to a reduced signal-to-noise ratio, i.e., model parameters smaller than in the centralized setting. This may explain why blind averaging seems to work for Softmax-SLP and SVMs with little noise (see Tbl. 2, 67x variant). Further, increasing the regularization parameter $\Lambda$ to help convergence can lead to poor accuracy of the converged model, e.g. for unbalanced datasets. For a detailed discussion, cf. Appx. B.

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

## A  EVALUATION: TRULY MANY USERS

The advantage of our method over DP-FL becomes especially evident when considering significantly more users (cf. Fig. 5), such as is common in distributed training via smartphones. Here, DP-guarantees of $\varepsilon \leq 5 \cdot 10^{-5}$ become plausible with at least $87\%$ prediction performance for a task like CIFAR-10. Alternatively, leveraging Cor. 9 we can consider a local DP scenario (with $\Upsilon = 50$) without a trusted aggregator, yielding an accuracy of $87\%$ for $\varepsilon = 1 \cdot 10^{-4}$. Starting from $\Upsilon \geq 2$, a user-level sensitivity (cf. Cor. 10) is in the evaluated setting mostly tighter than a data point dependent one; hence, the accuracy values are close to the local DP scenario.

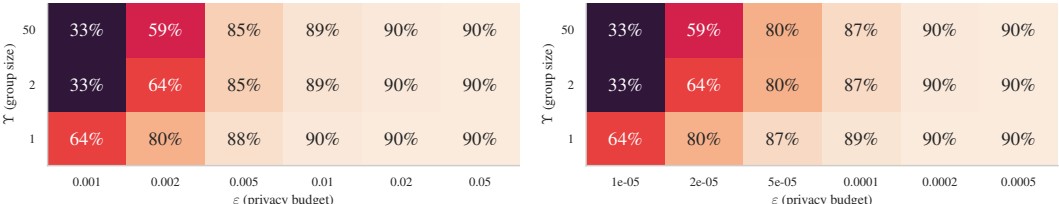

Figure 5: $(\varepsilon, \Upsilon)$-Heatmap for classification accuracy of Secure Distributed DP-Helmet (cf. Sec. 4) with learning algorithm DP_Softmax_SLP_SGD on CIFAR-10 dataset (left: $\delta = 10^{-10}$; right: $\delta = 10^{-12}$) with roughly 200,000 (left) and roughly 20,000,000 (right) users. We train 1,000 models on 50 data points each; to emulate having more users we rescale the $\varepsilon$-values ($\varepsilon' := 1000 \cdot \varepsilon \cdot \Upsilon / n_{users}$) to roughly reach the target number of users and report interpolated accuracy values. We extrapolate the privacy guarantees, due to the limited dataset size. Our accuracy values are pessimistic as we keep the accuracy numbers that we got from averaging 1,000 models. Actually taking the mean over roughly 200,000 or even roughly 20,000,000 users should provide better results. $\Upsilon < 50$ group privacy places trust in users as local aggregators whereas $\Upsilon = 50$ is comparable to local DP. Rescaling the $\varepsilon$-values only approximates the $\varepsilon$-guarantee we would get if we actually rescaled the noise scale by the target number of users. For $\Upsilon \geq 2$, a tighter group-privacy bound is possible (cf. Cor. 10); hence, the accuracy values are close to $\Upsilon = 50 = N$, where we protect the entire local dataset of a user.

## B  LIMITATIONS & DISCUSSION

**Distributional shifts between the public and sensitive datasets.**    For pre-training our models, we leverage contrastive learning. While very effective generally, contrastive learning is susceptible to performance loss if the shape of the sensitive data used to train the SVMs is significantly different from the shape of the initial public training data.

**Input Clipping.**    DP_SVM_SGD requires a norm bound on the input data as it directly influences the SVM training. In many pre-training methods like SimCLR no natural bound exists thus we have to artificially norm clip the input data. To provide a non-data-dependent clipping bound in CIFAR-10 data, we determined the clipping bound on the CIFAR-100 dataset (here: 34.854); its similar data distribution encompasses the output distribution of the pretraining reasonably well. Similarly, for CIFAR-100 data we determined the clipping bound based on the CIFAR-10 dataset (here: 34.157).

**Hyperparameter Search.**    In SVM_SGD, we deploy two performance-crucial hyperparameters: the regularization weight $\Lambda$ as well as the predictor radius $R$, both of which influence noise scaling. In the noise scaling subterm $c/\Lambda + R$, the maximal predictor radius is naturally significantly smaller than $c/\Lambda$ due to the regularization penalty. Thus, an imperfect $R$ resulting from a non-hyperparameter-tuned SVM does not have a large impact on the performance. Estimating the regularization weight for a fixed $\varepsilon$ from public data is called hyperparameter freeness in prior work (Iyengar et al., 2019). For other $\varepsilon$ values we can fit a (linear) curve on a smaller but related public dataset (proposed by Chaudhuri et al. (2011)) or synthetic data (proposed by AMP-NT (Iyengar et al., 2019)) as smaller $\varepsilon$ prefer higher regularization weights and vice versa.

**Blind averaging – Signal-to-noise ratio.**    For unfavorable datasets, we identify as a main limitation of blind averaging a reduced signal-to-noise ratio. In other words, the model is not as large as in the

centralized setting w.r.t. the sensitivity analysis. This effect would explain why our experiments show that with very little noise, blind averaging seems to work for Softmax-SLP and SVMs.

For SVMs, our formal characterization of the effect of blind averaging enables us to more precisely describe potential limitations of blind averaging for SVMs. In summary, we see two effects that reduce the signal-to-noise ratio. One effect comes from the requirement of the SVM training that the model with the smallest norm shall be found that satisfies the soft-margin constraints of the training data points. The local SVM training has fewer data points and, thus, fewer constraints. Hence, unfavorable local data sets will lead to a smaller model. Another effect comes from the averaging itself. Unfavorable local data sets can lead to local models that point in very different directions. When averaging these models, their norm naturally decreases as for any two vectors $a, b \in \mathbb{R}^n$ we have $0.5\|a + b\|_2 \leq 0.5(\|a\|_2 + \|b\|_2)$, and this discrepancy is larger the smaller the inner product is.

**Blind averaging – Unbalanced datasets.** Convergence holds if all data points are support vectors which implies a large margin, yet an SVM chooses roughly equally many support vectors per class: by the dual problem, we have the constraint $y^T \alpha = 0$ for labels $y_j \in \{-1, 1\}$ and dual coefficients $\alpha$. If we have a support vector inside the margin then $\alpha_j = \Lambda^{-1}$. Hence, enlarging the margin such that all data points are support vectors can lead to poor utility performance.

**Active attacks.** Active attackers may deviate from the protocol or send maliciously construed local models. If the secure summation protocol used is resilient against active adversaries and can still guarantee that only the sum of the inputs is leaked, then privacy is preserved. This follows from analyzing our algorithm for just the honest users and then leveraging the post-processing property of differential privacy. Secure summation protocols such as Bell et al. (2020) leak partial sums under active attacks and will diminish the privacy offered by our work against such adversaries as well.

## C EXTENDED PRELIMINARIES

### C.1 DIFFERENTIAL PRIVACY

To ease our analysis, we consider a randomized mechanism $M$ to be a function translating a database to a random variable over possible outputs. Running the mechanism then is reduced to sampling from the random variable. With that in mind, the standard definition of differential privacy looks as follows.

**Definition 15** ($\approx_{\varepsilon,\delta}$ relation). *Let $Obs$ be a set of observations, and $\mathrm{RV}(Obs)$ be the set of random variables over $Obs$, and $\mathcal{D}$ be the set of all databases. A randomized algorithm $M : \mathcal{D} \to \mathrm{RV}(Obs)$ for a pair of datasets $\mathcal{D}, \mathcal{D}'$, we write $M(D) \approx_{\varepsilon,\delta} M(D')$ if for all tests $S \subseteq Obs$ we have*

$$\Pr[M(D) \in S] \leq \exp(\varepsilon) \Pr[M(D') \in S] + \delta. \tag{1}$$

**Definition 16** (Differential Privacy). *Let $Obs$ be a set of observations, and $\mathrm{RV}(Obs)$ be the set of random variables over $Obs$, and $\mathcal{D}$ be the set of all databases. A randomized algorithm $M : \mathcal{D} \to \mathrm{RV}(Obs)$ for all pairs of databases $\mathcal{D}, \mathcal{D}'$ that differ in at most 1 element is a $(\varepsilon, \delta)$-DP mechanism if we have*

$$M(D) \approx_{\varepsilon,\delta} M(D'). \tag{2}$$

In the context of machine learning, the randomized algorithm represents the training procedure of a predictor. Our distinguishing element is one data record of the database.

**Computational Differential Privacy** Note that because of the secure summation, we technically require the computational version of differential privacy (Mironov et al., 2009), where the differential privacy guarantees are defined against computationally bounded attackers; the resulting increase in $\delta$ is negligible and arguments about computationally bounded attackers are omitted to simplify readability.

**Definition 17** (Computational $\approx_{\varepsilon,\delta}^c$ Differential Privacy). *Let $\mathcal{D}$ be the set of all databases and $\eta$ a security parameter. A randomized algorithm $M : \mathcal{D} \to \mathrm{RV}(Obs)$ for a pair of datasets $\mathcal{D}, \mathcal{D}'$, we write $M(D) \approx_{\varepsilon,\delta}^c M(D')$ if for any polynomial-time probabilistic attacker*

$$\Pr[A(M(D)) = 0] \leq \exp(\varepsilon) \Pr[A(M(D')) = 1] + \delta(\eta). \tag{3}$$

*For all pairs of databases $\mathcal{D}, \mathcal{D}'$ that differ in at most $1$ element $M$ is a computational $(\varepsilon, \delta(\eta))$-DP mechanism if we have*

$$M(D) \approx_{\varepsilon,\delta}^c M(D'). \tag{4}$$

## C.2 SECURE SUMMATION

Hiding intermediary local training results as well as ensuring their integrity is provided by an instance of secure multi-party computation (MPC) called secure summation (Bonawitz et al., 2017; Bell et al., 2020). It is targeted to comply with distributed summations across a huge number of parties. In fact, Bell et al. (2020) has a computational complexity for $n$ users on an $l$-sized input of $\mathcal{O}(\log^2 n + l \log n)$ for the client and $\mathcal{O}(n(\log^2 n + l \log n))$ for the server as well as a communication complexity of $\mathcal{O}(\log^2 n + l)$ for the client and $\mathcal{O}(n(\log n + l))$ for the server thus enabling an efficient run-through of roughly $10^9$ users without biasing towards computationally equipped users. Additionally, it offers resilience against client dropouts and colluding adversaries, both of which are substantial features for our distributed setting.

Before being able to formulate the security of the secure summation protocol, we need to a network execution against global network attacker that is active and adaptive. For the sake of self-containedness, we briefly present the notion of interactive machines and a sequential activation network execution. More general frameworks for such a setting include, e.g., the universal composability framework (Canetti, 2000).

We rely on the notion of interactive machines. For two interactive machines $X, Y$, we write $\langle X, Y \rangle$ for the interaction between $X$ and $Y$. We write $\langle X, Y \rangle = b$ to state that the machine $X$ terminates and outputs $b$.

**The network execution Real$_\pi$.** Next, we define network executions against global network attacker that is active and adaptive. Given a protocol $\pi$ with client and server code, we define an interactive machine Real$_\pi$ that lets each client party run the client code, lets the servers run the server code, and emulates a (sequential-activation-based) network execution, and interacts with another machine, called the attacker $\mathcal{A}$. The interaction is written as $\langle \mathcal{A}, \text{Real}_\pi \rangle$. Whenever within this network execution a party $B$ sends a message $m$ over the network to a party $C$, the interactive machine Real$_\pi$, sends this message $m$ to the attacker, activates the attacker, and waits for a response $m'$ from the attacker. Real$_\pi$ then lets this response $m'$ be delivered to party $C$, and activates party $C$. Moreover, the attacker $\mathcal{A}$ can send a dedicated message $(\texttt{compromise}, P)$ for compromising a party $P$ within the protocol execution. Whenever the attacker sends the message $(\texttt{compromise}, P)$ to the network execution Real$_\pi$, the network execution marks this party $P$ as compromised and sends the internal state of this party to the attacker $\mathcal{A}$. For each compromised party $P$, the attacker decides how $P$ acts. Formally, the network execution redirects each message $m$ that is sent to $P$ to the attacker $\mathcal{A}$ and awaits a response message $(m', P')$ from the attacker $\mathcal{A}$. Upon receiving the response $(m', P')$, the network execution Real$_\pi$ sends on behalf of $P$ the message $m'$ to the party $P'$.

For convenience, we write that a party $P$ *runs the client code of a protocol $\pi$ on input $m$* when the network execution runs for party $P$ the client code of $\pi$ on input $m$.

**Definition 18** (Secure Summation). *Let $\mathcal{F}(s_1, \ldots, s_n) := \sum_{i=1}^{n} s_i$. We say that $\pi_{SecSum}$ is secure summation if there is a probabilistic polynomial-time simulator $Sim_{\mathcal{F}}$ such that if a fraction of clients is corrupted ($C \subseteq \{U^{(1)}, \ldots, U^{(n)}\}$, $|C| = \gamma n$), $Real_{\pi_{SecSum}}(s_1, \ldots, s_n)$ is statistically indistinguishable from $Sim_{\mathcal{F}}(C, \mathcal{F}(s_1, \ldots, s_n))$, i.e., for an unbounded attacker $\mathcal{A}$ there is a negligible function $\nu$ such that*

$Advantage(\mathcal{A}) =$

$$|\Pr[langle \mathcal{A}, Real_{\pi_{SecAgg}}(s_1, \ldots, s_n)\rangle = 1] - \Pr[\langle \mathcal{A}, Sim_{\mathcal{F}}(C, \mathcal{F}(s_1, \ldots, s_n))\rangle = 1]| \leq \nu(\eta).$$

*Here, $Sim_{\mathcal{F}}$ is a potentially interactive simulator that only has access to the sum of all elements and the (sub-) set of corrupted clients. The adversary is unable to distinguish interactions and outputs of the simulator from those of the real protocol.*

The following theorem is proven for global network attackers that are passive and statically compromise parties. Formally, the theorem holds for all attackers $(\mathcal{A}', \mathcal{A}'')$ of the following form. $\mathcal{A}'$ internally runs $A''$ and ensures that only static compromisation is possible and that the attacker remains passive.

**Theorem 19** (Secure Aggregation $\pi_{SecAgg}$ in the semi-honest setting exists (Bell et al., 2020)). *Let $s_1, \ldots, s_n$ be the $d$-dimensional inputs of the clients $U^{(1)}, \ldots, U^{(n)}$. Let $\mathcal{F}$ be the ideal secure summation function: $\mathcal{F}(s_1, \ldots, s_n) := 1/n \sum_{i=1}^{n} s_i$. If secure authentication encryption schemes and authenticated key agreement protocol exist, the fraction of dropouts (i.e., clients that abort*

the protocol) is at most $\rho \in [0, 1]$, at most a $\gamma \in [0, 1]$ fraction of clients is corrupted ($C \subseteq \{ U^{(1)}, \ldots, U^{(n)} \}$, $|C| = \gamma n$), and the aggregator is honest-but-curious, there is a secure summation protocol $\pi_{SecAgg}$ for a central aggregator and $n$ clients that securely emulates $\mathcal{F}$ as in Def. 18.

### C.3   Pre-training to boost DP performance

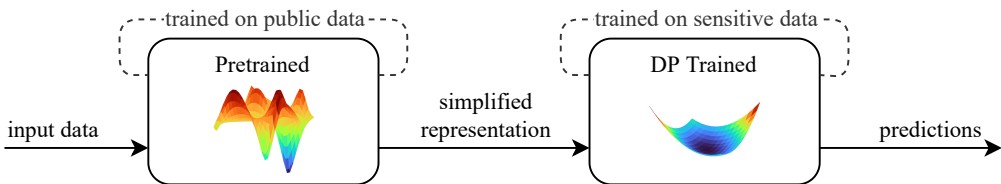

Figure 6: Pre-training: Schematic overview. Dashed lines denote data flow in the training phase and solid lines in the inference phase.

Recent work (Tramèr & Boneh, 2021; De et al., 2022) has shown that strong feature extractors (such as SimCLR (Chen et al., 2020a;b)), trained in an unsupervised manner, can be combined with simple learners to achieve strong utility-privacy tradeoffs for high-dimensional data sources like images. As a variation to transfer learning, it delineates a two-step process (cf. Fig. 6), where a simplified representation of the high-dimensional data is learned first before a tight privacy algorithm like DP_SVM_SGD conducts the prediction process on these simplified representations. For that, two data sources are compulsory: a public data source which is used to undertake the learning of a framework aimed to obtain pertinent simplified representations in addition to our sensitive data source that conducts the prediction process in a differentially private manner. Thereby the sensitive dataset is protected while strong expressiveness is assured through the use of the feature reduction network. Also note that a homogeneous data distribution of the public and the sensitive data is not necessarily required.

Recent work has shown that for several applications such representation reduction frameworks can be found, such as SimCLR for pictures, FaceNet for face images, UNet for segmentation, or GPT-3 for language data. Without loss of generality, we focus in this work on the unsupervised SimCLR feature reduction network (Chen et al., 2020a;b). SimCLR uses contrastive loss and image transformations to align the embeddings of similar images while keeping those of dissimilar images separate (Chen et al., 2020a). It is based upon a self-supervised training scheme called contrastive loss where no labeled data is required. Labelless data is especially useful as it exhibits possibilities to include large-scale datasets which would otherwise be unattainable due to the labeling efforts needed.

### C.4   Representer Theorem

With the representer theorem, we can completely describe a converged SVM $T(D)$ on $N$-sized dataset $D$ using a sum of the dual coefficients and the data points: $T(D) = \sum_{j=1}^{N} \alpha_j x_j$. The requirements for the representer theorem to hold are listed in the theorem below and include most notably an L2-regularized ERM objective.

**Theorem 20** (Representer theorem, cf. Argyriou et al. (2009) Lem 3, Thm 8). *Given a configuration $\zeta$, a local dataset $D^{(i)} := \{ (x_j, y_j) \}_{j=1}^{N} \subseteq \mathcal{H} \times \mathcal{Y}$ on a Hilbert space $\mathcal{H}$ with $\dim(\mathcal{H}) \geq 2$ and label space $\mathcal{Y}$, and a locally trained model of a learning algorithm $T$ such that there exists a solution that belongs to $\mathrm{span}(\{ x_j \}_{j=1}^{N})$, where $T(D^{(i)}, \xi, k) = \mathrm{argmin}_{f \in \mathcal{H}} E(\{ \langle f, x_j \rangle, y_j \}_{j=1}^{N}) + \Lambda\Omega(f)$ for some arbitrary error function $E \colon (\mathbb{R} \times \mathcal{Y})^N \mapsto \mathbb{R}$ and differentiable regularizer $\Omega \colon \mathcal{H} \mapsto \mathbb{R}$. Then $T$ admits a solution of the form $T(D^{(i)}, \xi, k) = \sum_{j=1}^{N} \alpha_i x_i$ for some $\alpha_i \in \mathbb{R}$ if and only if $\forall_{f \in \mathcal{H}} \Omega(f) = h(\langle f, f \rangle)$ with $h \colon \mathbb{R}_+ \mapsto \mathbb{R}$ as a non-descreasing function.*

In the case of SVM_SGD and Softmax_SLP_SGD, we have $\Omega = \|f\|^2$ which fulfills the requirements of the representer theorem since $h(z) = z$ is a linear function and the learning algorithm $T$ follows the definitions after convergence: $E(\{ \langle f, x_j \rangle, y_j \}_{j=1}^{N}) = \frac{1}{N} \sum_{(x,y) \in D^{(i)}} \ell_{\mathrm{hinge}}(y \langle f, x \rangle)$ is

the error function of SVM_SGD and $E(\{\langle f, x_j \rangle, y_j\}_{j=1}^N) = \frac{1}{N} \sum_{(x,y) \in D^{(i)}} \ell_{\text{softmax}}(y \langle f, x \rangle)$ the one of Softmax_SLP_SGD.

## C.5 DP_SVM_SGD

**Definition 21.** *The Huber loss according to Chaudhuri et al. (2011, Equation 7) is with a smoothness parameter $h$ defined as*

$$\ell_{huber}(h, z) := \begin{cases} 0 & \text{if } z > 1 + h \\ \frac{1}{4h}(1 + h - z)^2 & \text{if } |1 - z| \leq h \\ 1 - z & \text{if } z < 1 - h \end{cases}.$$

# D RELATED WORK

## D.1 PRIVACY-PRESERVING DISTRIBUTED MACHINE LEARNING

There is a rich body of literature about different differentially private distributed learning techniques that protect any individual data point (sometimes called distributed learning with global DP guarantees). One direction uses an untrusted central aggregator; users locally add noise to avoid leakage toward the aggregator. This method computationally scales well with the number of users. Another direction utilizes cryptographic protocols to jointly train a model without a central aggregator. This direction requires less noise for privacy, but the cryptographic protocols face scalability challenges.

For local noising, the most prominent and flexible approach is federated learning (McMahan et al., 2017) with DP-SGD approximation (Abadi et al., 2016) (DP-FL). DP-FL proposes each of the $n$ users locally train with the DP-SGD algorithm and share their local gradient updates with a central aggregator. This aggregator updates a global model with the average of the noisy local updates, leading to noise overhead in the order of $\sqrt{n}$.

This noise overhead can be completely avoided by PPDML protocols that rely on cryptographic methods to hide intermediary training updates from a central aggregator. There are several secure distributed learning methods that protect the contributions during training but do not come with privacy guarantees for the model such as DP: an attacker (e.g., a curious training party) can potentially extract information about the training data from the model. As we focus on differentially private distributed learning methods (PPDML in this paper), we will neglect those methods.

cpSGD (Agarwal et al., 2018) is a PPDML protocol that utilizes MPC methods to honestly generate noise and compute DP-SGD. While cpSGD provides the full flexibility of SGD, it does not scale to millions of users as it relies on expensive MPC methods. Another recent PPDML work (Truex et al., 2019) relies on a combination of MPC and DP methods. This work, however, also does not scale to millions of users.

Another line of research aims for the stronger privacy goal of protecting a user's entire input (called local DP) during distributed learning (Balle et al., 2020b; Girgis et al., 2021). Due to the strong privacy goal, federated learning with local DP tends to achieve weaker accuracy. With Cor. 9, evaluated in Fig. 5 in Appx. A, we show how Secure Distributed DP-Helmet achieves a comparable guarantee via group privacy: given enough users, any user can protect their entire dataset at once while we still reach good accuracy.

For DP training of SVMs, there exist other methods, such as objective perturbation and gradient perturbation. When performed under MPC-based distributed training, both methods would require a significantly higher number of MPC invocations; hence, they are unsuited for the goals of this work. Appx. D.2 discusses those approaches in detail.

## D.2 DIFFERENTIALLY PRIVATE EMPIRICAL RISK MINIMIZATION

On differentially private empirical risk minimization for convex loss functions (Chaudhuri et al., 2011), which is utilized in this work, the literature discusses three directions: output perturbation, objective perturbation, and gradient perturbation. Output perturbation (Chaudhuri et al., 2011; Wu et al., 2017) estimates a sensitivity on the final model without adding noise, and only in the end adds

noise that is calibrated to this sensitivity. We rely on output perturbation because it enables us to only have a single invocation of an MPC protocol at the end to merge the models while still achieving the same low sensitivity as if the model was trained at a trustworthy central party that collects all data points, trains a model and adds noise in the end.

Objective perturbation (Chaudhuri et al., 2011; Kifer et al., 2012; Iyengar et al., 2019; Bassily et al., 2019) adds noise to the objective function instead of adding noise to the final model. In principle, MPC could also be used to emulate the situation that a central party as above trains a model via objective perturbation. Yet, in that case, each party would have to synchronize with every other party far more often, as no party would be allowed to learn how exactly the objective function would be perturbed. That would result in far higher communication requirements.

Concerning gradient perturbation (Bassily et al., 2014; Wang et al., 2017; Feldman et al., 2018; Bassily et al., 2019; Feldman et al., 2020), recent work has shown tight privacy bounds. In order to achieve the same low degree of required noise as in a central setting, MPC could be utilized. Yet, for SGD also multiple rounds of communication would be needed as the privacy proof (for convex optimization) does not take into account that intermediary gradients are leaked. Hence, the entire differentially private SGD algorithm for convex optimization would have to be computed in MPC, similar to cpSGD (see above).

### D.3 TRUSTWORTHY DISTRIBUTED NOISE GENERATION.

One core requirement of SMPC-based distributed learning is honestly generated and unleakable noise as otherwise, our privacy guarantees would not hold anymore. There is a rich body of work on distributed noise generation (Moran et al., 2009; Dwork et al., 2006a; Kairouz et al., 2015b; 2021; Goryczka & Xiong, 2015). So far, however, no distributed noise generation protocol scales to millions of users. Thus, we use a simple, yet effective technique: we add enough noise if at least a fraction of them (say $t = 50\%$) are not colluding to violate privacy by sharing the noise they generate with each other.

## E EXPERIMENTAL SETUP

We leveraged 5-repeated 6-fold stratified cross-validation for all experiments unless stated differently. Privacy Accounting has been undertaken either by using the privacy bucket (Meiser & Mohammadi, 2018; Sommer et al., 2019) toolbox[2] or, for Gaussians without subsampling, with Sommer et al. (2019, Theorem 5) where both can be extended to multivariate Gaussians (see Appx. N). We note that with either of these tactics, $\varepsilon \in \mathcal{O}(|K|^{1/2})$. The $\delta$ parameter of differential privacy has been set to $\delta = 10^{-5}$ if not stated otherwise, which is for the CIFAR-10 and CIFAR-100 dataset always below $1/n$, where $n$ is the sum of the size of all local datasets.

Concerning computation resources, for our CIFAR-10 experiments, we trained 1000 DP_SVM_SGD with 50 data points each, which took 10 minutes on a machine with 2x *Intel Xeon Platinum 8168*, 24 Cores @2.7 GHz with an Nvidia A100 and allocated 16GB RAM.

For DP_SVM_SGD-based experiments, we utilize the strongly convex projected stochastic gradient descent algorithm (PSGD) as used by Wu et al. (2017). More specifically, we chose a batch size of 20, the Huber loss with a smoothness parameter $h = 0.1$. Furthermore for CIFAR-10, we chose a hypothesis space radius $R \in \{\, 0.04, 0.05, 0.06, 0.07, 0.08 \,\}$, a regularization parameter $\Lambda \in \{\, 10, 100, 200 \,\}$, and trained for 500 epochs; for the variant where we protect the whole local dataset, we have chosen a different $\Lambda \in \{\, 0.5, 1, 2, 5 \,\}$ and $R \in \{\, 0.06, 0.07 \,\}$. For CIFAR-100, we chose a chose a hypothesis space radius $R \in \{\, 0.04, 0.06, 0.08 \,\}$, a regularization parameter $\Lambda \in \{\, 3, 10, 30, 100 \,\}$, and trained for 150 epochs.

For DP_Softmax_SLP_SGD-based experiments, we utilize the strongly convex projected stochastic gradient descent algorithm (PSGD) as used by Wu et al. (2017) with a batch size of 20. Furthermore for CIFAR-10, we chose a hypothesis space radius $R \in \{\, 0.1, 0.4, 0.6, 1.0 \,\}$, a regularization parameter $\Lambda \in \{\, 1, 3, 10, 30 \,\}$, and trained for 150 epochs; for the variant where we protect the whole local dataset, we have chosen a different $\Lambda \in \{\, 0.5, 1 \,\}$ and $R \in \{\, 1, 3 \,\}$. For CIFAR-100,

---

[2]accessible at `https://github.com/sommerda/privacybuckets`, MIT license

we chose a chose the following hypothesis space radius and regularization parameter combinations: $(R, \Lambda) \in \{ (0.01, 100), (0.03, 30), (0.03, 100), (0.1, 10), (0.1, 30), (0.3, 3), (0.3, 10), (1, 1), (1, 3) \}$, and trained for 150 epochs.

For the strongly-biased non-iid experiments of Tbl. 2, we reported the results of the following hyperparameters: (CIFAR10, DP_SVM_SGD, regular & non-iid) $R = 0.06, \Lambda = 100$ for the dataset multiplier 1x and $R = 0.06, \Lambda = 10$ for the dataset multiplier 67x; (CIFAR10, DP_Softmax_SLP_SGD, regular) $R = 1.0, \Lambda = 1$ for both dataset multipliers 1x, 67x; (CIFAR10, DP_Softmax_SLP_SGD, non-iid) $R = 0.6, \Lambda = 3$ for both dataset multipliers 1x, 67x; (CIFAR100, DP_Softmax_SLP_SGD, regular) $R = 1.0, \Lambda = 3$ for dataset multiplier 1x and $R = 1.0, \Lambda = 1$ for both dataset multiplier 67x; (CIFAR100, DP_Softmax_SLP_SGD, non-iid) $R = 1.0, \Lambda = 3$ for both dataset multipliers 1x, 67x.

In every experiment, we chose for each parameter combination the best performing regularization parameter $\Lambda$ as well as $R$, i.e. those values that lead to the best mean accuracy. This is highly important, as the regularization parameter not only steers the utility but also the amount of noise needed to the effect where there is a sweet spot for each noise level where the amount of added noise is on the edge of still being bearable.

For the federated learning experiments, we utilized the *opacus*[3] PyTorch library (Yousefpour et al., 2021), which implements DP-SGD (Abadi et al., 2016). We loosely adapted our hyperparameters to the ones reported by Tramèr & Boneh (2021) who already evaluated DP-SGD on SimCLR's embeddings for the CIFAR-10 dataset. In detail, the neural network is a single-layer perceptron which has the following configuration: (CIFAR-10) 61,450 trainable parameters on a $6\,144\,\mathrm{d}$ input and $10\,\mathrm{d}$ output as well as (CIFAR-100) 614,500 trainable parameters on a $6,144\,\mathrm{d}$ input and $100\,\mathrm{d}$ output. The loss function is the categorical cross-entropy on a softmax activation function and training has been performed with stochastic gradient descent. Furthermore, we set the learning rate to $4$, the Poisson sample rate $q := {}^{1024}/_{50000}$ which in expectation samples a batch size of 1024, trained for 40 epochs, and norm-clipped the gradients with a clipping bound $c := 0.1$.

In the distributed training scenario, instead of running an end-to-end experiment with full MPC clients, we evaluate a functionally equivalent abstraction without cryptographic overhead. In our experiments, we randomly split the available data points among the users and emulated scenarios where not all data points were needed by taking the first training data points. However, the validation size remained constant. Moreover, for DP-SGD-based federated learning, we kept a constant batch size whenever enough data is available i.e. increased the sampling rate as follows: $q' := {}^{1024}/_{20000}$ for 20000, $q'' := {}^{1024}/_{5000}$ for 5000, and $q'' := {}^{1023}/_{1024}$ for 500 available data points $(|\mathcal{U}| \cdot N)$. For DP-SGD-based FL, we emulated a higher number of users by dividing the noise multiplier $\sigma$ by $|\mathcal{U}|^{1/2}$ to the benefit of DP-FL. The justification for dividing by $|\mathcal{U}|^{1/2}$ is that in FL the model performance is not expected to differ as the mean of the gradients of one user is the same as the mean of gradients from different users: SGD computes, just as FL, the mean of the gradients. Yet, the noise will increase by a factor of $|\mathcal{U}|^{1/2}$. Hence, we optimistically assume that everything stays the same, just the noise increases by a factor of $|\mathcal{U}|^{1/2}$.

## F  PRE-TRAINING VISUALISATION

## G  EXTENDED ABLATION STUDY (CENTRALIZED SETTING)

### G.1  SETUP OF THE ABLATION STUDY

For DP_SVM_SMO-based experiments, we used the *liblinear* (Fan et al., 2008) library via the Scikit-Learn method *LinearSVC*[4] for classification. *Liblinear* is a fast C++ implementation that uses the SVM-agnostic sequential minimal optimization (SMO) procedure. However, it does not offer a guaranteed and private convergence bound.

---

[3]accessible at `https://github.com/pytorch/opacus/`, Apache-2.0 license

[4]`https://scikit-learn.org/stable/modules/generated/sklearn.svm.LinearSVC.html`, BSD-3-Clause license

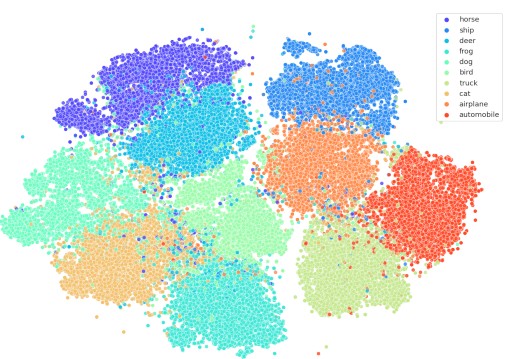

Figure 7: 2-d projection of the CIFAR-10 dataset via t-SNE (Van der Maaten & Hinton, 2008) with colored labels. Note that t-SNE is defined on the local neighborhood thus global patterns or structures may be arbitrary.

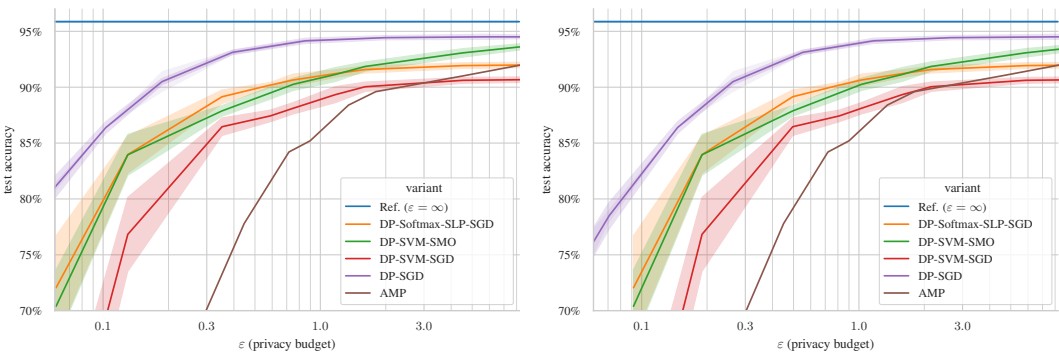

Figure 8: Classification accuracy compared to the privacy budget $\varepsilon$ of DP_SVM_SGD (cf. Sec. 3.1), DP_SVM_SMO where only the optima are perturbed, DP-SGD (1-layer perceptron) (Abadi et al., 2016), and AMP (SVM with objective perturbation) (Iyengar et al., 2019) on CIFAR-10 benchmark dataset (left: $\delta = 10^{-5}$, right: $\delta = 2 \cdot 10^{-8} \ll 1/\text{dataset\_size}$). For comparison, we report a non-private SVM baseline.

More specifically, we used the $L_2$-regularized hinge loss, an SMO convergence tolerance of $tol := 2 \cdot 10^{-12}$ with a maximum of 10,000 iterations which were seldom reached, and a logarithmically spaced inverse regularization parameter $C \in \{\{3, 6\} \cdot 10^{-8}, \{1, 2, 3, 6\} \cdot 10^{-7}, \{1, 2, 3, 6\} \cdot 10^{-6}, \{1, 2, 3, 6\} \cdot 10^{-5}, \{1, 2\} \cdot 10^{-4}\}$. To better fit with the *LinearSVC* implementation, the original loss function is rescaled by $1/\Lambda$ and $C$ is set to $1/\Lambda \cdot n$ with $n$ as the number of data points. Furthermore, for distributed DP_SVM_SMO training we extended the range of the hyperparameter $C$ – whenever appropriate – up to $3 \cdot 10^{-3}$ which becomes relevant in a scenario with many users and few data points per user. Similar to DP_SVM_SGD-based experiment, the best performing regularization parameter $C$ was selected for each parameter combination.

The non-private reference baseline uses a linear SVM optimized via SMO with the hinge loss and an inverse regularization parameter $C = 2$ (best performing of $C \in \{\leq 5 \cdot 10^{-5}, 0.5, 1, 2\}$).

For the ablation study, we also included the Approximate Minima Perturbation (AMP) algorithm[5] (Iyengar et al., 2019) which resembles an instance of objective perturbation. There, we used a (80–20)-train-test split with 10 repeats and the following hyperparameters: $L \in \{0.1, 1.0, 34.854\}$, eps_frac $\in \{.9, .95, .98, .99\}$, eps_out_frac $\in \{.001, .01, .1, .5\}$. We selected ($L = 1$, $eps\_out\_frac = 0.001$, $eps\_frac = 0.99$) as a good performing parameter combination for AMP. For better performance, we resembled the GPU-capable *bfgs_minimize* from the Tensorflow Probability package. To

---

[5]reference implementation by the authors: `https://github.com/sunblaze-ucb/dpml-benchmark`, MIT license

provide better privacy guarantees, we leveraged the results of Kairouz et al. (2015a); Murtagh & Vadhan (2016) for tighter composition bounds on arbitrary DP mechanisms.

## G.2 RESULTS OF THE ABLATION STUDY

For the extended ablation study, we considered the centralized setting (only 1 user) and compare different algorithms as well as different values for the privacy parameter $\delta$. The results are depicted in Fig. 8 and display five algorithms: firstly, the differentially private Support Vector Machine with SGD-based training DP_SVM_SGD (cf. Sec. 3.1), secondly, the differentially private Softmax-activated single-layer perceptron with SGD-based training DP_Softmax_SLP_SGD (cf. Sec. 4.1), thirdly, a similar differentially private SVM but with SMO-based training which does not offer a guaranteed and private convergence bound, fourthly, differentially private Stochastic Gradient descent (DP-SGD) (Abadi et al., 2016) applied on a 1-layer perceptron with the cross-entropy loss, and fifthly, approximate minima perturbation (AMP) (Iyengar et al., 2019) which is based upon an SVM with objective perturbation. Note that, only DP_SVM_SMO, DP_SVM_SGD, and DP_Softmax_SLP_SGD have an output sensitivity and are thus suited for this efficient Secure Distributed DP-Helmet scheme.

While all algorithms come close to the non-private baseline with rising privacy budgets $\varepsilon$, we observe that although DP-SGD performs best, DP_SVM_SMO and DP_Softmax_SLP_SGD come considerably close, DP_SVM_SGD has a disadvantage above DP_SVM_SMO of about a factor of 2, and AMP a disadvantage of about a factor of $4$. We suspect that DP-SGD is able to outperform the variants other than DP_Softmax_SLP_SGD as it directly optimizes for the multi-class objective via the cross-entropy loss while others are only able to simulate it via the one-vs-rest (ovr) SVM training scheme. Additionally, DP-SGD has a noise-correcting property from its iterative noise application. The inherently multi-class DP_Softmax_SLP_SGD performs better than ovr-based DP_SVM_SGD indicating that a joint learning of all classes can boost performance. DP_Softmax_SLP_SGD additionally has a privacy advantage as it does not need to rely on sequential composition as it has an output sensitivity for all classes which is another factor that can lead to the boost of DP_Softmax_SLP_SGD above DP_SVM_SGD. Although DP_SVM_SMO has an output sensitivity as well and renders better than DP_SVM_SGD, it does not offer a privacy guarantee when convergence is not reached. In the case of AMP, we have an inherent disadvantage of about a factor of $3$ due to an unknown output distribution, and thus bad composition results in the multi-class SVM. Here, the privacy budget of AMP roughly scales linearly with the number of classes.

For DP-SGD, DP_SVM_SGD, DP_Softmax_SLP_SGD, and DP_SVM_SMO, Fig. 8 shows that a smaller and considerably more secure privacy parameter $\delta \ll 1/dataset\_size$ is supported although reflecting on the reported privacy budget $\varepsilon$.

## H PROOF OF LEM. 6

We recall Lem. 6:

**Lemma 6** (Sensitivity of Secure Distributed DP-Helmet). *For a configuration $\zeta$, Secure Distributed DP-Helmet$(\zeta)$ of Alg. 2 without noise, $avg_i(T(D^{(i)}))$, has a sensitivity of $s \cdot 1/|\mathcal{U}|$ for each class $k \in K$.*

*Proof.* Without loss of generality, we consider one arbitrary class $k \in K$. We know that $T$ is an $s$-sensitivity bounded algorithm thus

$$s = \max_{D_0^{(i)} \sim D_1^{(i)}} \left| T(D_0^{(i)}, \xi, k) - T(D_1^{(i)}, \xi, k) \right| \tag{5}$$

with $D_0^{(i)}$ and $D_1^{(i)}$ as 1-neighboring datasets. For instance, for $T = $ SVM_SGD we have $s = \frac{2(c+R\Lambda)}{N\Lambda}$ (cf. Lem. 2).

By Alg. 2, we take the average of multiple local models, i.e. $avg_i(T(D^{(i)})) = \frac{1}{|\mathcal{U}|} \sum_{i=1}^{|\mathcal{U}|} T(D^{(i)}, \xi, K)$. The challenge element – i.e. the element that differs between $D_0^{(i)}$ and $D_1^{(i)}$ – is only contained in one of the $|\mathcal{U}|$ models. By the application of the parallel composition theorem,

we know that the sensitivity reduces to

$$\max_{D_0^{(i)} \sim D_1^{(i)}, \forall i=0,\dots,|\mathcal{U}|} \left| \frac{1}{|\mathcal{U}|} \sum_{i=1}^{|\mathcal{U}|} T(D_0^{(i)}, \xi, k) - \frac{1}{|\mathcal{U}|} \sum_{i=1}^{|\mathcal{U}|} T(D_1^{(i)}, \xi, k) \right| = s \cdot \frac{1}{|\mathcal{U}|}.$$

Hence, the constant $1/|\mathcal{U}|$ factor reduces the sensitivity by a factor of $1/|\mathcal{U}|$. $\qquad\square$

## I    PROOF OF LEM. 7

We recall Lem. 7:

**Lemma 7.** *For configuration $\zeta$ and noise scale $\tilde{\sigma}$: $\frac{1}{|\mathcal{U}|} \sum_{i=1}^{|\mathcal{U}|} \mathcal{N}(0, (\tilde{\sigma} \cdot 1/\sqrt{|\mathcal{U}|})^2) = \mathcal{N}(0, (\tilde{\sigma} \cdot 1/|\mathcal{U}|)^2)$.*

*Proof.* We have to show that

$$\frac{1}{|U|} \sum_{i=1}^{|U|} \mathcal{N}(0, (\tilde{\sigma} \cdot \frac{1}{\sqrt{|\mathcal{U}|}})^2) = \mathcal{N}(0, (\tilde{\sigma} \cdot \frac{1}{|\mathcal{U}|})^2). \tag{6}$$

It can be shown that the sum of normally distributed random variables behaves as follows: Let $X \sim \mathcal{N}(\mu_X, \sigma_X^2)$ and $Y \sim \mathcal{N}(\mu_Y, \sigma_Y^2)$ two independent normally-distributed random variables, then their sum $Z = X + Y$ equals $Z \sim \mathcal{N}(\mu_X + \mu_Y, \sigma_X^2 + \sigma_Y^2)$ in the expectation.

Thus, in this case, we have

$$\frac{1}{|U|} \sum_{i=1}^{|U|} \mathcal{N}(0, (\tilde{\sigma} \cdot \frac{1}{\sqrt{|\mathcal{U}|}})^2) = \frac{|U|}{|U|} \mathcal{N}(0, |U| \cdot (\tilde{\sigma} \cdot 1/\sqrt{|\mathcal{U}|})^2) = \frac{1}{|U|} \mathcal{N}(0, \tilde{\sigma}^2).$$

As the normal distribution belongs to the location-scale family, we get $\mathcal{N}(0, (\tilde{\sigma} \cdot 1/|\mathcal{U}|)^2)$. $\qquad\square$

## J    PROOF OF THM. 8

We state the full version of Thm. 8:

**Theorem 8** (Main Theorem, full). *For a configuration $\zeta$ as in Def. 5, a maximum fraction of dropouts $\rho \in [0, 1]$, and a maximum fraction of corrupted clients $\gamma \in [0, 1]$. Assume that secure summation $\pi_{SecSum}$ exists as in Def. 18.*

*Then Secure Distributed DP-Helmet($\zeta$) (cf. Alg. 2) satisfies computational $(\varepsilon, \delta + \nu_1)$-DP with $\varepsilon \geq \sqrt{2 \ln 1.25/(\delta/|K|)} \cdot |K| \cdot 1/\sigma$, for $\nu_1 := (1 + \exp(\varepsilon)) \cdot \nu(\eta)$ and a function $\nu$ negligible in the security parameter $\eta$ used in $\pi_{SecSum}$.*

*Proof.* We first show $(\varepsilon, \delta)$-DP for a variant $M_1$ of Secure Distributed DP-Helmet that uses the ideal summation protocol $\mathcal{F}$ instead of $\pi_{SecSum}$. We conclude that for Secure Distributed DP-Helmet (abbreviated as $M_2$) which uses the real secure summation protocol $\pi_{SecSum}$ for some negligible function $\nu_1$ $(\varepsilon, \delta + \nu_1)$-DP holds.

Recall that we assume at least $t \cdot |\mathcal{U}|$ many honest users. As we solely rely on the honest $t \cdot |\mathcal{U}|$ to contribute correctly distributed noise to the learning algorithm $T$, we have for each class similar to Lem. 7

$$\frac{1}{|\mathcal{U}|} \sum_{i=1}^{t \cdot |\mathcal{U}|} \mathcal{N}(0, (\tilde{\sigma} \cdot \frac{1}{\sqrt{|\mathcal{U}|}})^2) = \sum_{i=1}^{t \cdot |\mathcal{U}|} \mathcal{N}(0, (\tilde{\sigma} \cdot \frac{1}{|\mathcal{U}|\sqrt{|\mathcal{U}|}})^2)$$

$$= \mathcal{N}(0, (\tilde{\sigma} \cdot \frac{\sqrt{t \cdot |\mathcal{U}|}}{|\mathcal{U}|\sqrt{|\mathcal{U}|}})^2) = \mathcal{N}(0, (\tilde{\sigma} \cdot \frac{\sqrt{t}}{|\mathcal{U}|})^2).$$

Hence, we scale the noise parameter $\tilde{\sigma}$ with $1/\sqrt{t}$ and get

$$\frac{1}{|\mathcal{U}|} \sum_{i=1}^{t|\mathcal{U}|} \mathcal{N}(0, (\tilde{\sigma} \cdot \frac{1}{\sqrt{t}} \cdot \frac{1}{\sqrt{|\mathcal{U}|}})^2) = \mathcal{N}(0, (\tilde{\sigma} \cdot \frac{1}{|\mathcal{U}|})^2).$$

By Lem. 6, Lem. 7, and Lem. 3, we know that $M_1$ satisfies $(\varepsilon, \delta)$-DP (with the parameters as described above).

Considering an unbounded attacker $\mathcal{A}$, we know that for any pair of neighboring data sets $D, D'$ the following holds

$$\Pr\left[\mathcal{A}\left(\mathcal{M}_1(D)\right) = 1\right] \leq \exp(\varepsilon) \Pr\left[\mathcal{A}\left(\mathcal{M}_1(D')\right) = 1\right] + \delta$$

If $\pi_{SecSum}$ is a secure summation protocol, there is a negligible function $\nu$ such that for any neighboring data sets $D, D'$ (differing in at most one element) the following holds w.l.o.g.:

$$\Pr\left[\mathcal{A}\left(\mathcal{M}_2(D)\right) = 1\right] - \nu(\eta) \leq \Pr\left[\mathcal{A}\left(Sim_{\mathcal{F}}(\mathcal{M}_1(D))\right) = 1\right]. \tag{7}$$

For the attacker $\mathcal{A}'$ that first applies $Sim$ and then $\mathcal{A}$, we get:

$$\Pr\left[\mathcal{A}\left(\mathcal{M}_2(D)\right) = 1\right] - \nu(\eta) \leq \exp(\varepsilon) \Pr\left[\mathcal{A}\left(Sim_{\mathcal{F}}(\mathcal{M}_1(D'))\right) = 1\right] + \delta \tag{8}$$

$$\leq \exp(\varepsilon)\left(\Pr\left[\mathcal{A}\left(\mathcal{M}_2(D')\right) = 1\right] + \nu(\eta)\right) + \delta \tag{9}$$

thus we have

$$\Pr\left[\mathcal{A}\left(\mathcal{M}_2(D)\right) = 1\right] \leq \exp(\varepsilon) \Pr\left[\mathcal{A}\left(\mathcal{M}_2(D')\right) = 1\right] + \delta + (1 + \exp(\varepsilon)) \cdot \nu(\eta). \tag{10}$$

From a similar argumentation it follows that

$$\Pr\left[\mathcal{A}\left(\mathcal{M}_2(D')\right) = 1\right] \leq \exp(\varepsilon) \Pr\left[\mathcal{A}\left(\mathcal{M}_2(D)\right) = 1\right] + \delta + (1 + \exp(\varepsilon)) \cdot \nu(\eta) \tag{11}$$

holds.

Hence, with $\nu_1 := (1 + \exp(\varepsilon)) \cdot \nu(\eta)$ the mechanism Secure Distributed DP-Helmet mechanism $\mathcal{M}_2$ which uses $\pi_{SecSum}$ is $(\varepsilon, \delta + \nu_1)$-DP. As $\nu$ is negligible and $\varepsilon$ is constant, $\nu_1$ is negligible as well.

$\square$

**Corollary 22.** *Given a configuration $\zeta$, a maximum fraction of dropouts $\rho \in [0, 1]$, and a maximum fraction of corrupted clients $\gamma \in [0, 1]$, if secure authentication encryption schemes and authenticated key agreement protocol exist, then Secure Distributed DP-Helmet($\zeta$) (cf. Alg. 2) instantiated with $\pi_{SecSum} = \pi_{SecAgg}$ (Bell et al., 2020) satisfies computational $(\varepsilon, \delta + \nu_1)$-DP with $\varepsilon \geq \sqrt{2 \ln 1.25/(\delta/|K|)} \cdot |K| \cdot 1/\sigma$, for $\nu_1 := (1 + \exp(\varepsilon)) \cdot \nu(\eta)$ and a function $\nu$ negligible in the security parameter $\eta$ used in secure summation.*

This follows directly from Thm. 8, as by Thm. 19, we know that $\pi_{SecAgg}(s_1, \ldots, s_n)$ securely emulates $\mathcal{F}$ (w.r.t. an unbounded attacker).

## K    PROOF OF COR. 10

We recall Cor. 10:

**Corollary 10** (User-level sensitivity). *Given a learning algorithm $T$, we say that $T$ is $R$-norm bounded if for any input dataset $D$ with $N = |D|$, any hyperparameter $\xi$, and all classes $k \in K$, $\|T(D, \xi, k)\| \leq R$. Any $R$-norm bounded learning algorithm $T$ has a sensitivity $s = 2R$. In particular, $T + \mathcal{N}(0, (\sigma \cdot s \cdot I_d)^2)$ satisfies $(\varepsilon, \delta)$, $\Upsilon$-group differential privacy with $\Upsilon = N$ and $\varepsilon \geq \sqrt{2 \ln 1.25/(\delta/|K|)} \cdot |K| \cdot 1/\sigma$, where $\mathcal{N}(0, (\sigma \cdot s \cdot I_d)^2)$ is spherical multivariate Gaussian noise and $\sigma$ a noise multiplier.*

*Proof.* We know that the sensitivity of the learning algorithm $T$ is defined as $s = \max_{D \sim D'} \|T(D, \xi, k) - T(D', \xi, k)\|$ for $\Upsilon$-neighboring datasets $D, D'$. Thus, in our case we have $s = 2R$ since any $T(\_, \xi, k) \in [-R, R]$. As this holds independent on the dataset and by Lem. 3 and by Lem. 24, we can protect any arbitrary number of data points per user, i.e. we have $\Upsilon$-group DP. $\square$

## L  NON-INTERACTIVE BLIND AVERAGE

**Corollary 23** (Averaged Representer theorem). *Given a configuration $\zeta$, if on $D^{(i)}$ locally learning algorithm $T$ admits a solution of the form $T(D^{(i)}, \xi, k) = \sum_{j=1}^{N} \alpha_j^{(i)} x_j^{(i)}$ (cf. Thm. 20) then the average of these locally trained models $avg_i(T(D^{(i)}))$ admits a solution of the form $T(\mho, \xi, k) = \frac{1}{|\mathcal{U}|} \sum_{i=1}^{|\mathcal{U}|} \sum_{j=1}^{N} \alpha_j^{(i)} x_j^{(i)}$.*

*Proof.*

$$avg_i(T(D^{(i)})) = \frac{1}{|\mathcal{U}|} \sum_{i=1}^{|\mathcal{U}|} T(D^{(i)}) = \frac{1}{|\mathcal{U}|} \sum_{i=1}^{|\mathcal{U}|} \sum_{j=1}^{N} \alpha_j^{(i)} x_j^{(i)}.$$

$\square$

### L.1  PROOF OF LEM. 13

We recall Lem. 13:

**Lemma 13** (Support Vectors of averaged SVM). *Given a configuration $\zeta$ as in Def. 5, a locally trained model of learning algorithm $T$, where $T(D^{(i)}, \xi, k) = \operatorname{argmin}_f \frac{1}{N} \sum_{(x,y) \in D^{(i)}} \max(0, 1 - f^T xy \cdot (1[y = k] - 1[y \neq k])) + \Lambda \|f\|^2$ comprises a hinge-loss linear SVM, has the support vectors $V^{(i)} = \left\{ (x,y) \in D^{(i)} \mid T(D^{(i)})^T xy \leq \|T(D^{(i)})\|^{-1} \right\}$. Then, the average of these locally trained models $avg_i(T(D^{(i)}))$ has the support vectors $V = \bigcup_{i=1}^{|\mathcal{U}|} V^{(i)}$.*

*Proof.* A learning problem that is based on a hinge-loss SVM fulfills the representer theorem requirements due to the L2-regularized ERM objective function. In fact, if a data point $x_j$ is a support vector, i.e. $x_j \in V$, then after successful training its corresponding $\alpha_j$ is restricted by $0 < \alpha_j \leq \Lambda \wedge y_j = 1$ or $0 > \alpha_j \geq -\Lambda \wedge y_j = -1$, or $\alpha_j = 0$ (Ma & Ng, 2020, Equation 28-30). Thus, we denote $V^{(i)} = \left\{ x_j^{(i)} \in D^{(i)} \mid \alpha_j^{(i)} \neq 0 \right\}$. By Cor. 23, we have that the average of locally trained models $avg_i(T(D^{(i)})) = \frac{1}{|\mathcal{U}|} \sum_{i=1}^{|\mathcal{U}|} \sum_{j=1}^{N} \alpha_j^{(i)} x_j^{(i)}$. Since the local datasets are disjoint we simplify $\frac{1}{|\mathcal{U}|} \sum_{i=1}^{|\mathcal{U}|} \sum_{j=1}^{N} \alpha_j^{(i)} x_j^{(i)} = \frac{1}{|\mathcal{U}|} \sum_{j=1}^{|\mho|} \alpha_j x_j$ for the combined local datasets $\mho = \bigcup_{i=1}^{|\mathcal{U}|} D^{(i)}$ and a flattened $\alpha = \begin{bmatrix} \alpha_1^{(1)} & \dots & \alpha_N^{(1)} & \alpha_1^{(2)} & \dots & \alpha_N^{(|\mathcal{U}|)} \end{bmatrix}$. A model which is represented by $\frac{1}{|\mathcal{U}|} \sum_{j=1}^{|\mho|} \alpha_j x_j$ has the support vectors $V = \{ x_j \in \mho \mid \alpha_j \neq 0 \} = \bigcup_{i=1}^{|\mathcal{U}|} V^{(i)}$, as the support vector characteristic is uniquely determined by $\alpha$ and each local $\alpha_j^{(i)}$ is element of $\alpha$ and responsible for the same data point. $\square$

### L.2  PROOF OF THM. 14

We recall Thm. 14:

**Theorem 14** (Averaging locally trained SVM converges to a global SVM). *Given a configuration $\zeta$ as in Def. 5, there exists a regularization parameter $\Lambda$ such that the average of locally trained models $avg_i(T(D^{(i)}))$ with a hinge-loss linear SVM as objective function $\mathcal{J}$ trained with projected subgradient descent using weighted averaging (PGDWA), $T = HINGE\_SVM\_PGDWA$, converges with the number of local iterations $M$ to the best model for the combined local datasets $\mho$, i.e.*

$$\mathbb{E}\left[ \mathcal{J}(avg_i(HINGE\_SVM\_PGDWA(D^{(i)})), \mho, \_) - \inf_f \mathcal{J}(f, \mho, \_) \right] \in \mathcal{O}(1/M).$$

*Proof.* First (1), we show that there exists a regularization parameter $\Lambda$ for which the converged global model equals the average of the converged locally trained models: $T(\mho) = avg_i(T(D^{(i)}))$. Second (2), we show that both the global and the local models converge with rate $\mathcal{O}(1/M)$.

Note that we assume that each data point $x_j$ is structured as $[1, x_{j,1}, \dots, x_{j,p}]$ to include the intercept. We also denote the flattened $\alpha^{(\text{avg\_loc})} = \begin{bmatrix} \alpha_1^{(1)} & \dots & \alpha_N^{(1)} & \alpha_1^{(2)} & \dots & \alpha_N^{(|\mathcal{U}|)} \end{bmatrix}$ as the dual coefficients of the averaged local SVM and $\alpha^{(\text{glob})}$ as the dual coefficients of the global SVM.

(1) By Lem. 13 we know for the combined local datasets $\mho = \bigcup_{i=1}^{|\mathcal{U}|} D^{(i)}$ that

$$\text{avg}_i(T(D^{(i)})) = \frac{1}{|\mathcal{U}|N} \sum_{j=1}^{|\mho|} \alpha_j^{(\text{avg\_loc})} x_j = \frac{1}{|\mho|} \sum_{j=1}^{|\mho|} \alpha_j^{(\text{avg\_loc})} x_j.$$

Note that we assume a scaled parameter per local SVM: $T(D^{(i)}) = \frac{1}{N} \sum_{j=1}^N \alpha_j x_j$. Without this assumption, we would not average the local SVMs but instead compute their sum.

For the global model, we write by the representer theorem

$$T(\mho) = \frac{1}{|\mho|} \sum_{j=1}^{|\mho|} \alpha_j^{(\text{glob})} x_j.$$

Thus, by parameter comparison we have that $T(\mho) = \text{avg}_i(T(D^{(i)}))$ if $\forall_j \, \alpha_j^{(\text{glob})} = \alpha_j^{(\text{avg\_loc})}$. By the characteristic of a hinge-loss linear SVM, we know that any $\alpha_j$ has the value $\alpha_j = \Lambda y_j$ if a data point is a support vector inside the margin (Ma & Ng, 2020, Equation 28-30). Hence, $\forall_j \, \alpha_j^{(\text{glob})} = \alpha_j^{(\text{avg\_loc})}$ if the margin is large enough that for both SVMs all data points are inside the margin. Since the margin of a hinge-loss linear SVM is the inverse of the parameter norm, $\|T(\mho)\|^{-1}$, and the parameter norm gets smaller with an increased regularization parameter $\Lambda$ by the definition of the objective function $\frac{1}{N} \sum_{(x,y) \in D^{(i)}} \max(0, 1 - y f^T x) + \Lambda \|f\|^2$, we derive that there exists a regularization parameter $\Lambda$ which is large enough s.t. all data points are within the margin.

(2) By Lacoste-Julien et al. (2012), we know that a hinge-loss linear SVM converges to the optima with rate $\mathcal{O}(M^{-1})$, if we use projected subgradient descent using weighted averaging (PGDWA) as an optimization algorithm, i.e.

$$\mathbb{E}\big[\mathcal{J}(\text{avg}_i(\text{HINGE\_SVM\_PGDWA}(D^{(i)})), \mho, \_) - \inf_f \mathcal{J}(f, \mho, \_)\big] \in \mathcal{O}(1/M).$$

$\square$

## M  GROUP PRIVACY REDUCTION OF MULTIVARIATE GAUSSIAN

**Lemma 24.** *Let* $\text{pdf}_{\mathcal{N}(A,B)}[x]$ *denote the probability density function of the multivariate Gaussian distribution with location and scale parameters* $A, B$ *which is evaluated on an atomic event* $x$. *For any atomic event* $x$, *any covariance matrix* $\Sigma$, *any group size* $k \in \mathbb{N}$, *and any mean* $\mu$, *we get*

$$\frac{\text{pdf}_{\mathcal{N}(0,k^2\Sigma)}[x]}{\text{pdf}_{\mathcal{N}(\mu,k^2\Sigma)}[x]} = \frac{\text{pdf}_{\mathcal{N}(0,\Sigma)}[x/k]}{\text{pdf}_{\mathcal{N}(\mu/k,\Sigma)}[x/k]}. \tag{12}$$

*Proof.*

$$\frac{\text{pdf}_{\mathcal{N}(0,k^2\Sigma)}[x]}{\text{pdf}_{\mathcal{N}(\mu,k^2\Sigma)}[x]} = \frac{\frac{1}{det(2\pi k^2 \Sigma)} \exp(-\frac{1}{2} x^T k^2 \Sigma^{-1} x)}{\frac{1}{det(2\pi k^2 \Sigma)} \exp(-\frac{1}{2} \underbrace{(x-\mu)^T k^2 \Sigma^{-1} (x-\mu)}_{=x^T k^2 \Sigma^{-1} x - \mu^T k^2 \Sigma^{-1} x - x^T k^2 \Sigma^{-1} \mu + \mu^T k^2 \Sigma^{-1} \mu})} \tag{13}$$

$$= \exp(-\frac{1}{2}(-\mu^T k^2 \Sigma^{-1} x - x^T k^2 \Sigma^{-1} \mu + \mu^T k^2 \Sigma^{-1} \mu)) \tag{14}$$

$$= \exp(-\frac{1}{2} k^2 \cdot (-\mu^T \Sigma^{-1} x - x^T \Sigma^{-1} \mu + \mu^T \Sigma^{-1} \mu)) \tag{15}$$

for $\mu_1 := \mu/k$

$$= \exp(-\frac{1}{2} \cdot k(-\mu_1^T \Sigma^{-1} x - x^T \Sigma^{-1} \mu_1 + \mu_1^T \Sigma^{-1} \mu_1/k)) \tag{16}$$

for $x_1 := x/k$

$$= \exp(-\frac{1}{2} \cdot (-\mu_1^T \Sigma^{-1} x_1 - x_1^T \Sigma^{-1} \mu_1 + \mu_1^T \Sigma^{-1} \mu_1)) \tag{17}$$

$$= \exp(-\frac{1}{2} \cdot (-\mu_1^T \Sigma^{-1} x_1 - x_1^T \Sigma^{-1} \mu_1 + \mu_1^T \Sigma^{-1} \mu_1)) \tag{18}$$

$$= \frac{\frac{1}{det(2\pi\Sigma)} \exp(-\frac{1}{2} x_1^T \Sigma^{-1} x_1)}{\frac{1}{det(2\pi\Sigma)} \exp(-\frac{1}{2}(x_1 - \mu_1)^T k^2 \Sigma^{-1}(x_1 - \mu_1))} \tag{19}$$

$$= \frac{\text{pdf}_{\mathcal{N}(0,\Sigma)}[x/k]}{\text{pdf}_{\mathcal{N}(\mu/k,\Sigma)}[x/k]} \tag{20}$$

$\square$

As the Gaussian distribution belongs to the location-scale family, Lem. 24 directly implies that the $(\varepsilon, \delta)$-DP guarantees of using $\mathcal{N}(0, k^2 \Sigma)$ noise for sensitivity $k$ and using $\mathcal{N}(0, \Sigma)$ for sensitivity 1 are the same.

## N    REPRESENTING MULTIVARIATE GAUSSIANS AS UNIVARIATE GAUSSIANS

For the sake of completeness, we rephrase a proof that we first saw in Abadi et al. (2016) that argues that sometimes the multivariate Gaussian mechanism can be reduced to the univariate Gaussian mechanism.

**Lemma 25.** *Let* $\text{pdf}_{\mathcal{N}(\mu,\text{diag}(\sigma^2))}$ *denote the probability density function of a multivariate ($p \geq 1$) spherical Gaussian distribution with location and scale parameters* $\mu \in \mathbb{R}^p, \sigma \in \mathbb{R}_+^p$. *Let* $M_{gauss,p,q}$ *be the $p$ dimensional Gaussian mechanism* $D \mapsto q(D) + \mathcal{N}(0, \sigma^2 \cdot I_p)$ *for* $\sigma^2 > 0$ *of a function* $q : \mathcal{D} \to \mathcal{R}^p$, *where* $\mathcal{D}$ *is the set of datasets. Then, for any $p \geq 1$, if $q$ is $s$-sensitivity-bounded, then for any $p \geq 1$, there is another $s$-sensitivity-bounded function* $q' : \mathcal{D} \to \mathcal{R}$ *such that the following holds: for all $\varepsilon \geq 0, \delta \in [0,1]$ if* $M_{gauss,1,q'}$ *satisfies $(\varepsilon, \delta)$-ADP, then $M_{gauss,p,q}$ satisfies $(\varepsilon, \delta)$-ADP.*

*Proof.* First observe that for any $s$-sensitivity-bounded function $q''$, two adjacent inputs $D, D'$ (differing in one element) with $\|q''(D) - q''(D')\|_2 = s$ are worst-case inputs. As a spherical Gaussian distribution (covariance matrix $\Sigma = \sigma^2 \cdot I_{p \times n}$) is rotation invariant, there is a rotation such that the difference only occurs in one dimension and has length $s$. Hence, it suffices to analyze a univariate Gaussian distribution with sensitivity $s$. Hence, the privacy loss distribution of both mechanisms (for the worst-case inputs) is the same. As a result, for all $\varepsilon \geq 0, \delta \in [0,1]$ (i.e. the privacy profile is the same) if $(\varepsilon, \delta)$-ADP holds for the univariate Gaussian mechanism it also holds for the multivariate Gaussian mechanism. $\square$

## O    STRONG CONVEXITY OF CE LOSS

**Theorem 26.** *Let* $\mathcal{J}$ *denote the objective function* $\mathcal{J}(f, D) := \frac{\Lambda}{2} \sum_{k=1}^K (f^T f)_k + \frac{1}{N} \sum_{(x,y) \in D} \mathcal{L}_{CE}(y, f^T x)$ *with the cross-entropy loss* $\mathcal{L}_{CE}(y, z) := -\sum_{k=1}^K y_k \log \frac{\exp(z_k)}{\sum_{j=1}^K \exp(z_j)}$ *and parameters* $f \in \mathbb{R}^{d+1,K}$, *dataset $D$ where $(x, y) \in D$ with data points $x \in \mathbb{R}^{d+1}$ structured as* $\begin{bmatrix} 1 & x_1 & \dots & x_d \end{bmatrix}$ *and labels* $y \in \{0, 1\}^K$, *number of classes $K$, and regularization parameter $\Lambda$.* $\mathcal{J}$ *is $\Lambda$-strongly convex.*

*Proof.* $\mathcal{J}$ is $\mu$-strongly convex if $\mathcal{J} - \frac{\mu}{2} \|f\|$ is convex. In our case, with $\mu = \Lambda$, it remains to be shown show that the cross entropy loss $\mathcal{L}_{CE}(y, z)$ is convex, since a linear layer like $f^T x$ represents an affine map which preserves convexity (Bertsekas, 2009).

It is known that the cross entropy loss is convex by a simple argumentation: If the Hessian is positive semidefinite $\mathcal{L}_{CE}(y, z) \succeq 0$ then $\mathcal{L}_{CE}$ is convex. By the Gershgorin's circle theorem is a symmetric diagonally dominant matrix positive semi-definite if the diagonals are non-real.

Since the second derivative of the cross-entropy loss is $\frac{\partial^2}{\partial z_p \partial z_q} \mathcal{L}_{\text{CE}} = s_p(1_{[p=q]} - s_q)$ for the softmax probabilities $s_p = \frac{\exp(z_p)}{\sum_{j=1}^K \exp(z_j)}$, we conclude that the diagonals are non-negative since $s_p(1 - s_p)$ for $0 \le s_p \le 1$ is always non-negative. The Hessian is diagonal dominant, if for every row $p$ the absolute value of the diagonal entry is larger or equal the sum of the absolute values of all other row entries. In our case, we have

$$\forall_p \, |s_p(1-s_q)| \ge \sum_{q=1, q\neq p}^K |s_p(-s_q)| \iff \forall_p \, (1-s_q) \ge \sum_{q=1, q\neq p}^K s_q \iff \forall_p \, (1-s_q) \ge (1-s_p).$$

$\square$

## P    LIPSCHITZNESS OF CE LOSS

**Theorem 27.** *Let $\mathcal{J}$ denote the objective function $\mathcal{J}(f, D) := \frac{\Lambda}{2} \sum_{k=1}^K (f^T f)_k + \frac{1}{N} \sum_{(x,y)\in D} \mathcal{L}_{CE}(y, f^T x)$ with the cross-entropy loss $\mathcal{L}_{CE}(y, z) := -\sum_{k=1}^K y_k \log \frac{\exp(z_k)}{\sum_{j=1}^K \exp(z_j)}$ and parameters $f \in \mathbb{R}^{d+1, K}$, dataset $D$ where $(x, y) \in D$ with data points $x \in \mathbb{R}^{d+1}$ structured as $\begin{bmatrix} 1 & x_1 & \dots & x_d \end{bmatrix}$ and labels $y \in \{0, 1\}^K$, number of classes $K$, and regularization parameter $\Lambda$. $\mathcal{J}$ is L-Lipschitz with $L = \Lambda R + \sqrt{2}c$ where $\|x\| \le c$ and $\|f\| \le R$.*

*Proof.* In the following, we abbreviate $d' := d + 1$, flatten $f \in \mathbb{R}^{d'K}$ and notate $z := (x, y)$.

The Lipschitz continuity is defined as:

$$\sup_{z\in D, f, f'} \frac{\|\mathcal{J}(f, z) - \mathcal{J}(f', z)\|}{\|f - f'\|} \le L.$$

We first (1) show

$$\sup_{z\in D, f, f'} \frac{\|\mathcal{J}(f, z) - \mathcal{J}(f', z)\|}{\|f - f'\|} \le \sup_{z\in D, f} \|\nabla_f \mathcal{J}(f, z)\|$$

using the mean value theorem and subsequently (2) bound $\sup_{z\in D, f} \|\nabla_f \mathcal{J}(f, z)\| \le L$.

(1) Recall that the multivariate mean value theorem states that for some function $g \colon G \mapsto \mathbb{R}$ on an open subset $G \in \mathbb{R}^n$, some $x, y \in G$ and some $c \in [0, 1]$, we have

$$g(y) - g(x) = \langle \nabla g((1 - c)x + cy), y - x \rangle.$$

In our case, we write

$$\sup_{z\in D, f, f'} \frac{\|\mathcal{J}(f, z) - \mathcal{J}(f', z)\|}{\|f - f'\|}$$

by the multivariate mean value theorem for some $c \in [0, 1]$

$$= \sup_{z\in D, f, f'} \frac{|\langle \nabla \mathcal{J}((1-c)f' - cf, z), f - f'\rangle|}{\|f - f'\|}$$

for $f'' := (1 - c)f' - cf$ and by the Cauchy-Schwarz inequality $|\langle \nabla_{f''} \mathcal{J}(f'', z), f - f'\rangle| \le \|\nabla_{f''} \mathcal{J}(f'', z)\| \cdot \|f - f'\|$

$$\le \sup_{z\in D, f''} \|\nabla_{f''} \mathcal{J}(f'', z)\|.$$

(2) We know that for $1 \leq j \leq d', 1 \leq p \leq K$ the partial derivative of $\mathcal{J}$ is $\frac{\partial}{\partial f_p} \mathcal{J}(f, (x, y)) = \Lambda f_{lp} + x_l \cdot (s_p - 1_{[y=p]})$ with $s_p := \frac{\exp(f_p x)}{\sum_{j=1}^{K} \exp(f_j x)}$. Thus, we have

$$\|\nabla_f \mathcal{J}(f, z)\| = \sqrt{\sum_{lp=1}^{d'K} \left( \Lambda f_{lp} + x_l (s_p - 1_{[y=p]}) \right)^2}$$

$$= \sqrt{\sum_{lp=1}^{d'K} \left( \Lambda^2 f_{lp}^2 + 2\Lambda f_{lp} x_l (s_p - 1_{[y=p]}) + x_l^2 (s_p - 1_{[y=p]})^2 \right)}$$

$$= \sqrt{\Lambda^2 \|f\|^2 + 2\Lambda \sum_{l=1}^{d'} x_l \sum_{p=1}^{K} f_{lp}(s_p - 1_{[y=p]}) + \sum_{l=1}^{d'} x_l^2 \sum_{p=1}^{K} (s_p - 1_{[y=p]})^2}$$

due to the Cauchy-Schwarz inequality, we have $\sum_{p=1}^{K} f_{lp}(s_p - 1_{[y=p]}) \leq \sqrt{\sum_{p=1}^{K} f_{lp}^2} \sqrt{\sum_{p=1}^{K} (s_p - 1_{[y=p]})^2}$ and $\sum_{l=1}^{d'} x_l \sqrt{\sum_{p=1}^{K} f_{lp}^2} \leq \sqrt{\sum_{l=1}^{d'} x_l^2} \sqrt{\sum_{lp=1}^{d'K} f_{lp}^2} = \|x\|^2 \|f\|^2$

$$\leq \sqrt{\Lambda^2 \|f\|^2 + 2\Lambda \|x\| \|f\| \sqrt{\sum_{p=1}^{K} (s_p - 1_{[y=p]})^2} + (\sum_{l=1}^{d'} x_l^2)(\sum_{p=1}^{K} (s_p - 1_{[y=p]})^2)}$$

since $\max_{s_1,\dots,s_K} \left\{ (s_p - 1)^2 + \sum_{q=1,q\neq p}^{K} s_q^2 \mid \sum_{k=1}^{K} s_k = 1 \wedge s_k \geq 0 \, \forall_k \right\} = 2$ with $s_q = 1 \wedge s_p = 0 \bigwedge_{k=1,k\neq q}^{K} s_k = 0$ where $q \neq p$

$$\leq \sqrt{\Lambda^2 \|f\|^2 + 2\sqrt{2}\Lambda \|x\| \|f\| + 2 \|x\|^2} = \Lambda \|f\| + \sqrt{2} \|x\|$$

Thus, with $\|x\| \leq c, \|f\| \leq R$ we conclude that

$$\sup_{z \in D, f, f'} \frac{\|\mathcal{J}(f, z) - \mathcal{J}(f', z)\|}{\|f - f'\|} \leq \sup_{z \in D, f} \|\nabla_f \mathcal{J}(f, z)\| \leq \Lambda R + \sqrt{2}c = L$$

$\square$

## Q  SMOOTHNESS OF CE LOSS

**Theorem 28.** *Let $\mathcal{J}$ denote the objective function $\mathcal{J}(f, D) := \frac{\Lambda}{2} \sum_{k=1}^{K} (f^T f)_k + \frac{1}{N} \sum_{(x,y) \in D} \mathcal{L}_{CE}(y, f^T x)$ with the cross-entropy loss $\mathcal{L}_{CE}(y, z) := -\sum_{k=1}^{K} y_k \log \frac{\exp(z_k)}{\sum_{j=1}^{K} \exp(z_j)}$ and parameters $f \in \mathbb{R}^{d+1,K}$, dataset $D$ where $(x, y) \in D$ with data points $x \in \mathbb{R}^{d+1}$ structured as $[1 \quad x_1 \quad \dots \quad x_d]$ and labels $y \in \{0, 1\}^K$, number of classes $K$, and regularization parameter $\Lambda$. $\mathcal{J}$ is $\beta$-smooth with $\beta = \sqrt{(d+1)K\Lambda^2 + 0.5(\Lambda + c^2)^2}$ where $\|x\| \leq c$.*

*Proof.* In the following, we abbreviate $d' := d + 1$, flatten $f \in \mathbb{R}^{d'K}$ and notate $z := (x, y)$.

$\beta$-Smoothness is defined as:

$$\sup_{z \in D, f, f'} \frac{\|\nabla_f \mathcal{J}(f, z) - \nabla_{f'} \mathcal{J}(f', z)\|}{\|f - f'\|} \leq \beta.$$

We first (1) show

$$\sup_{z \in D, f, f'} \frac{\|\nabla_f \mathcal{J}(f, z) - \nabla_{f'} \mathcal{J}(f', z)\|}{\|f - f'\|} \leq \sup_{z \in D, f} \|\mathbf{H}_f(\mathcal{J}(f, z))\|$$

using the mean value theorem and subsequently (2) bound $\sup_{z \in D, f} \|\mathbf{H}_f(\mathcal{J}(f, z))\| \leq \beta$.

(1) Recall that the multivariate mean value theorem states that for some function $g\colon G \mapsto \mathbb{R}$ on an open subset $G \in \mathbb{R}^n$, some $x, y \in G$ and some $c \in [0, 1]$, we have

$$g(y) - g(x) = \langle \nabla g((1-c)x + cy), y - x \rangle.$$

In our case, we write

$$\sup_{z \in D, f, f'} \frac{\left\| \nabla_f \mathcal{J}(f, z) - \nabla_{f'} \mathcal{J}(f', z) \right\|}{\|f - f'\|}$$

$$= \sup_{z \in D, f, f'} \frac{\sqrt{\sum_{i=0}^{d'K} (\nabla_{f_i} \mathcal{J}(f, z) - \nabla_{f_i'} \mathcal{J}(f', z))^2}}{\|f - f'\|}$$

by the multivariate mean value theorem for some $c \in [0, 1]$ and $g_i(f, z) := \nabla_{f_i} \mathcal{J}(f, z)$

$$= \sup_{z \in D, f, f'} \frac{\sqrt{\sum_{i=0}^{d'K} \langle \nabla g_i((1-c)f' - cf, z), f - f' \rangle^2}}{\|f - f'\|}$$

for $f'' := (1-c)f' - cf$ and by the Cauchy-Schwarz inequality $|\langle \nabla g_i(f'', z), f - f' \rangle|^2 \leq \|\nabla g_i(f'', z)\|^2 \cdot \|f - f'\|^2$

$$\leq \sup_{z \in D, f''} \sqrt{\sum_{i=0}^{d'K} \sum_{j=0}^{d'K} (\nabla^2_{f_i'', f_j''} \mathcal{J}(f'', z))^2}$$

$$= \sup_{z \in D, f} \|\mathbf{H}_f(\mathcal{J}(f, z))\|.$$

(2) We know that with $1 \leq l \leq d', 1 \leq p \leq K$ the first-order partial derivative of $\mathcal{J}$ is $\frac{\partial}{\partial f_{lp}} \mathcal{J}(f, (x, y)) = \Lambda f_{lp} + x_l \cdot (s_p - 1_{[y=p]})$ with $s_p := \frac{\exp(f_p^T x)}{\sum_{i=1}^K \exp(f_i^T x)}$.

With $1 \leq j \leq d', 1 \leq q \leq K$ we know that the second-order partial derivative of $\mathcal{J}$ is $\frac{\partial^2}{\partial f_{lp} \partial f_{jq}} \mathcal{J}(f, (x, y)) = 1_{[lp=jq]} \cdot \Lambda + x_l \cdot x_j \cdot s_p(1_{[p=q]} - s_q)$. Thus, we have

$$\|\mathbf{H}_f(\mathcal{J}(f, z))\| = \sqrt{\sum_{lp=1}^{d'K} \sum_{jq=1}^{d'K} \left(1_{[lp=jq]} \cdot \Lambda + x_l x_j s_p (1_{[p=q]} - s_q)\right)^2}$$

$$= \sqrt{\sum_{lp=1}^{d'K} \left((\Lambda + x_l^2 s_p(1 - s_p))^2 + \sum_{\substack{jq=1 \\ j \neq l}}^{d'K} x_l^2 x_j^2 s_p^2 (1 - s_p)^2 + \sum_{\substack{jq=1 \\ j \neq l \\ q \neq p}}^{d'K} x_l^2 x_j^2 s_p^2 s_q^2\right)}$$

$$= \sqrt{\sum_{lp=1}^{d'K} \left((\Lambda + x_l^2 s_p(1 - s_p))^2 + x_l^2 s_p^2 \sum_{\substack{j=1 \\ j \neq l}}^{d'} \left(x_j^2 (1 - s_p)^2 + x_j^2 \sum_{\substack{q=1 \\ q \neq p}}^K s_q^2\right)\right)}$$

since we have $\max_{s_1, \ldots, s_K} \left\{ \sum_{q=1, q \neq p}^K s_q^2 \mid \sum_{q=1, q \neq p}^K s_q = 1 - s_p \land s_i \geq 0 \, \forall_i \right\} = (1 - s_p)^2$ due to the maximal L2-distance given a bounded L1-distance is the maximal L2-distance in one dimension, we conclude

$$= \sqrt{\sum_{lp=1}^{d'K} \left(\Lambda^2 + 2\Lambda x_l^2 s_p(1 - s_p) + x_l^4 s_p^2(1 - s_p)^2 + 2 x_l^2 s_p^2(1 - s_p)^2 \sum_{\substack{j=1 \\ j \neq l}}^{d'} x_j^2\right)}$$

$$\leq \sqrt{d'K\Lambda^2 + \sum_{lp=1}^{d'K} x_l^2 s_p(1 - s_p) \left(2\Lambda + 2 x_l^2 s_p(1 - s_p) + 2 s_p(1 - s_p) \sum_{\substack{j=1 \\ j \neq l}}^{d'} x_j^2\right)}$$

$$= \sqrt{d'K\Lambda^2 + 2 \sum_{lp=1}^{d'K} x_l^2 s_p(1 - s_p) \left(\Lambda + s_p(1 - s_p) \|x\|^2\right)}$$

$$\leq \sqrt{d'K\Lambda^2 + 2 \|x\|^2 \sum_{l=1}^{d'} x_l^2 \sum_{p=1}^K s_p(1 - s_p)(\Lambda \|x\|^{-2} + s_p)}$$

following Lem. 29 (presented and shown below) we simplify with $C := \Lambda \|x\|^{-2}$: $\sum_{p=1}^{K} s_p (1 - s_p)(C + s_p) \leq 0.25(C + 1)^2$

$$\leq \sqrt{d' K \Lambda^2 + 0.5 \|x\|^4 (\Lambda \|x\|^{-2} + 1)^2} = \sqrt{d' K \Lambda^2 + 0.5(\Lambda + \|x\|^2)^2}.$$

Thus, with $\|x\| \leq c$ we conclude that

$$\sup_{z \in D, f, f'} \frac{\|\nabla_f \mathcal{J}(f, z) - \nabla_{f'} \mathcal{J}(f', z)\|}{\|f - f'\|} \leq \sup_{z \in D, f} \|\mathbf{H}_f(\mathcal{J}(f, z))\| \leq \sqrt{d' K \Lambda^2 + 0.5(\Lambda + c^2)^2} = \beta.$$

$\square$

**Lemma 29.** *Let* $\{s_p\}_{p=1}^{K}$ *denote probabilities such that* $\sum_{p=1}^{K} s_p = 1$, *and* $C \in \mathbb{R}_+$ *a constant, then we have*

$$\max_{\{s_p\}_{p=1}^{K}} \left\{ \sum_{p=1}^{K} s_p (1 - s_p)(C + s_p) \mid \sum_{p=1}^{K} s_p = 1 \wedge s_p \geq 0 \, \forall_p \right\} \leq 0.25(C + 1)^2$$

*with* $(s_p = \frac{1}{k} \wedge s_{p'} = 0) \, \forall_{p \in \cup_{i=1}^{k} P_i, p' \in \cup_{i=k+1}^{K} P_i, P \in \mathrm{Sym}(K)}$, *i.e. for some arbitrary but fixed dimensions* $k : 1 \leq k \leq K$, *the solution has* $k$-*times* $s_p = \frac{1}{k}$ *and* $(K - k)$-*times* $s_p = 0$.

*Proof.* We show this Lemma as follows: First, we use the Karush–Kuhn–Tucker (KKT) conditions to find the $s_p$'s which maximize the maximization term. Thereby, we obtain a set of four solution candidates where we encode all $s_p$'s in closed form and introduce two new variables $k, j$ which serve as a solution counter. Second, we insert the solution candidates into the maximization term and show that the result is always bounded by $0.25(C + 1)^2$ by calculating the optimal front across all possible values of the solution counters $k, j$.

Let $f(s) := \sum_{p=1}^{K} s_p (1 - s_p)(C + s_p)$ denote the function to maximize, $h(s) := \sum_{p=1}^{K} s_p - 1$ the equality constraint, and $g_p(s) := -s_p, \forall_p$ the inequality constraints. To find the constrained maximum, we maximize the Lagrangian function $\mathcal{L}_{agrange}(s) = f(s) + \mu_p g_p(s) + \lambda h(s)$ with $\mu_p, \lambda$ as slack variables. This suffices since $s_p$ does not have unbounded border cases: the only valid configuration of all $s_p$'s is on a hyperplane ($\sum_p s_p = 1$) bounded in all dimensions ($s_p \geq 0$). Using the slack variable $\mu_p$, we already cover whether its corresponding $s_p$ is on the border ($\mu_p > 0$) or not ($\mu = 0$). Following the KKT conditions, the following conditions have to hold for the maximum:

(1) Stationarity: $\nabla_{s_p} \mathcal{L}_{agrange}(s) = C + 2s_p - 2C s_p - 3s_p^2 + \mu_p - \lambda = 0, \forall_p$

(2) Primal feasibility: $h(s) = 0$ and $g_p(s) \leq 0, \forall_p$

(3) Dual feasibility: $\mu_p \geq 0, \forall_p$

(4) Complementary slackness: $\mu_p g_p(s) = 0, \forall_p$

Informally, it suffices for the solution of the KKT conditions to analyze the cases where $s_p > 0, \forall_{1 \leq p \leq k}$ for all fixed number of dimensions $k : 1 \leq k \leq K$ since if $s_p = 0$ then we have already proved the same result for one less dimension.

Formally and without loss of generality[6], we show for all fixed numbers of dimensions $k : 1 \leq k \leq K$ that for the solution of the KKT conditions it suffices to analyze the cases where $s_p > 0, \forall_{1 \leq p \leq k}$. For the induction base case ($k = 1$ dimensional), we have $s_1 > 0$ and thus by condition (4) $\mu_1 = 0$. If and only if $s_1 = 1$, we satisfy conditions (2) and (1) with $\lambda = -C - 1$. With $s_1 = 0$ we would not be able to satisfy the equality constraint of condition (2), i.e. '$s_1 = 1$'.
For the $k \mapsto k + 1$ induction case, we know that $s_p > 0, \forall_{1 \leq p \leq k}$. If $s_{p+1} > 0$, by the induction hypothesis we know that $\forall_{1 \leq p \leq k+1}, s_p = 0$. If $s_{p+1} = 0$ then by conditions (3) and (4) we have $\mu_{p+1} > 0$ and thus by condition (1), $\mu_{p+1} = \lambda - C$. Inserting $s_{p+1} = 0, \mu_{p+1} = \lambda - C$ into

---

[6]The same argumentation holds for situations where the dimensions are permuted.

conditions (1) to (4), we obtain the same set of equations and inequalities as for the $k$-dimensional case which already holds by the induction hypothesis.

We solve the KKT conditions (1) to (4) as follows: First, we solve the system of equations of condition (1) for $s_p$ via the quadratic formula:

$$s_p^{\pm} = \frac{-(2-2C) \pm \sqrt{(2-2C)^2 - 4(-3)(C-\lambda)}}{2(-3)} = \frac{1}{3} \cdot \left( \pm \sqrt{C^2 + C - 3\lambda + 1} - C + 1 \right). \quad (21)$$

Second, we plug $s_p^{\pm}$ into the equality constraint '$h(s) = 0$' of condition (2) and solve for $\lambda$ which gives us for some solution counter $j \in \mathbb{N}, 0 \leq j \leq k$ with $2j \neq k$:

$$h(s^{\pm}) = 0$$
$$\iff \left( \sum_{i=1}^{j} s_i^+ \right) + \left( \sum_{i=j+1}^{k} s_i^- \right) = 1$$
$$\iff j \left( \sqrt{C^2 + C - 3\lambda + 1} - C + 1 \right) + (k-j) \left( -\sqrt{C^2 + C - 3\lambda + 1} - C + 1 \right) = 3$$
$$\iff (2j-k) \sqrt{C^2 + C - 3\lambda + 1} = Ck - k + 3$$
$$\Rightarrow C^2 + C - 3\lambda + 1 = \frac{(Ck-k+3)^2}{(2j-k)^2}$$
$$\iff \lambda = \frac{(2j-k)^2(C^2+C+1) - (Ck-k+3)^2}{3(2j-k)^2}.$$

The solution counter $j$ quantifies how often we plug the 'positive' variant of $s_p^{\pm}$ into $h(s^{\pm})$:

$$s^{\pm} := \begin{bmatrix} s_1^+ & \cdots & s_j^+ & s_{j+1}^- & \cdots & s_k^- \end{bmatrix}$$

or any permutation of the dimensions of $s^{\pm}$.

Note that at $2j = k$, we have a special case and by the equality constraint '$h(s) = 0$' of condition (2)

$$h(s^{\pm}) = 0 \wedge 2j = k \iff \left( \sum_{p=1}^{\frac{k}{2}} s_p^+ \right) + \left( \sum_{p=\frac{k}{2}+1}^{k} s_p^- \right) = 1 \iff k(1-C) = 3 \iff C = \frac{k-3}{k}.$$

Thus, at $2j = k, C = \frac{k-3}{k}$ we simplify the solution in Eq. (21) to

$$s_p^{\pm, \, C = k - 3/k} = \frac{1}{3} \cdot ( \pm \underbrace{\sqrt{\frac{(k-3)^2}{k^2} + \frac{k-3}{k} - 3\lambda + 1}}_{=: \, 3Q} - \frac{k-3}{k} + 1) = \pm Q + \frac{1}{k}.$$

If we now insert $s_p^{\pm, \, C = k - 3/k}$ into $f(\cdot)$ and maximize for all remaining variables, we find the maximum at

$$\max_{k,j,\lambda} \left\{ f(s_p^{\pm, \, C=k-3/k}) \mid 2j = k \wedge C = \frac{k-3}{k} \wedge s_p^{\pm, \, C=k-3/k} \geq 0 \right\}$$
$$\leq \max_{k,j,\lambda} \left\{ \sum_{p=1}^{\frac{k}{2}} (\frac{1}{k} + Q)(1 - (\frac{1}{k} + Q))(C + (\frac{1}{k} + Q)) \right.$$
$$\left. + \sum_{p=\frac{k}{2}+1}^{k} (\frac{1}{k} - Q)(1 - (\frac{1}{k} - Q))(C + (\frac{1}{k} - Q)) \mid 2j = k \wedge C = \frac{k-3}{k} \right\}$$
$$= \max_{k} \left\{ \frac{k}{2} \frac{1}{k} (1 - \frac{1}{k})(\frac{k-3}{k} + \frac{1}{k}) + \frac{k}{2} \frac{1}{k} (1 - \frac{1}{k})(\frac{k-3}{k} + \frac{1}{k}) \right\}$$
$$= \max_{k} \left\{ (1 - \frac{1}{k})(\frac{k-3}{k} + \frac{1}{k}) \right\} = \max_{k} \left\{ \frac{k-2}{k} - \frac{k-2}{k^2} \right\} = \max_{k} \left\{ \underbrace{1 - \frac{3}{k} + \frac{2}{k^2}}_{\leq 0.25(\frac{k-3}{k}+1)^2 = 1 - \frac{3}{k} + \frac{9}{4k^2}} \right\}.$$

Thus, at $2j = k$, $\mathcal{L}_{\text{agrange}}$ is maximal at $C = \frac{k-3}{k}$ which is always strictly below the maximum we will show in this lemma if $C = \frac{k-3}{k}$. In the following, we continue the proof for $2j \neq k$.

Third, by plugging $\lambda$ into Eq. (21) which is derived from the system of equations in condition (1) and solving for $s_p$ we obtain the following two solution candidates for $2j \neq k$

$$s_p^{(+,-)} = \frac{1}{3} \left( \pm \sqrt{C^2 + C - 3\frac{(2j-k)^2(C^2+C+1)-(Ck-k+3)^2}{3(2j-k)^2} + 1} - C + 1 \right)$$
$$= \frac{1}{3} \left( 1 - C \pm \frac{Ck-k+3}{2j-k} \right) = \frac{(2j-k)(1-C) \pm (Ck-k+3)}{3(2j-k)} = \frac{-2Cj+Ck+2j-k \pm (Ck-k+3)}{6j-3k}$$
$$s_p^{(+)} = \frac{-2(k-j)C + 2(k-j) - 3}{6(k-j) - 3k}, s_p^{(-)} = \frac{-2jC + 2j - 3}{6j - 3k}.$$

Observe that if we replace $\tilde{j} := k - j$ in $s_p^{(+)}$ we get $s_p^{(-)}$ with $\tilde{j}$ instead of $j$. To abbreviate, we write

$$s_p^{(j')} = \frac{-2j'C + 2j' - 3}{6j' - 3k}$$

for $j' \in \{\, j, k - j \,\}$. Because of the similar structure of $s_p^{(j)}$ and $s_p^{(k-j)}$, restricting $j$ by $0 \le 2j < k$ suffices since we would otherwise count the same maximum twice. With $s_p^{(j')}$ as our solution candidate, the equality constraint '$h(s) = 0$' in condition (2) holds when we have $(k - j)$ times $s_p^{(j)}$ and $j$ times $s_p^{(k-j)}$:

$$s^{sol} := \begin{bmatrix} s_1^{(k-j)} & \cdots & s_j^{(k-j)} & s_{j+1}^{(j)} & \cdots & s_k^{(j)} \end{bmatrix}$$

or any permutations of the dimensions of $s^{sol}$. This goes by construction of $s^{\pm}$ where the solution counter $j$ quantifies how often we plug in $s_p^{(+)}$ into $h(s^{(+,-)})$.

We next compute the second partial derivative test to determine for which parameters the solution candidate $s^{sol}$ is a local maximum or minimum: We have a maximum if the Hessian of $\mathcal{L}_{agrange}$ is positive definite and a minimum if the Hessian of $\mathcal{L}_{agrange}$ is negative definite. In our case, the second partial derivatives of $\mathcal{L}_{agrange}$ are $\nabla_{s_p}^2 \mathcal{L}_{agrange}(s) = 2 - 2C - 6s$ and $\nabla_{s_p} \nabla_{s_q} \mathcal{L}_{agrange}(s) = 0$ with $p \ne q$. Thus, we have a diagonal Hessian matrix. Hence, if $2 - 2C - 6s^{sol} < \mathbf{0}$ we have a maximum and if $2 - 2C - 6s^{sol} > \mathbf{0}$ we have a minimum. Because of the second partial derivative test, we also know that if the Hessian has both positive and negative eigenvalues then we have a saddle point. This holds in our case when we have both positive and negative values on the diagonals of the Hessian, i.e. for some $p$ we have $2 - 2C - 6s_p^{sol} < 0$ and for some $q$ we have $2 - 2C - 6s_q^{sol} > 0$. Furthermore, if we have a zero eigenvalue this test is indecisive.

We rearrange the maximum condition for any entry of $s^{sol}$ (here: $s_p^{(j')}$) as follows:

$$2 - 2C - 6\frac{-2j'C + 2j' - 3}{6j' - 3k} < 0 \iff \begin{cases} kC - k + 3 > 0 & \text{if } 0 \le 2j' < k \\ kC - k + 3 < 0 & \text{if } 2j' > k \end{cases}$$

$$\iff \begin{cases} C > \frac{k-3}{k} & \text{if } 0 \le 2j' < k \\ C < \frac{k-3}{k} & \text{if } 2j' > k \end{cases}.$$

Similarly, we rearrange the minimum condition, such that

$$2 - 2C - 6\frac{-2j'C + 2j' - 3}{6j' - 3k} > 0 \iff \begin{cases} C < \frac{k-3}{k} & \text{if } 0 \le 2j' < k \\ C > \frac{k-3}{k} & \text{if } 2j' > k \end{cases}.$$

Recall that at this point we only consider $2j \ne k$. We now distinguish three cases for the second partial derivative test for the vector $s^{sol}$: $C < \frac{k-3}{k}$, $C > \frac{k-3}{k}$, $C = \frac{k-3}{k}$.

At $C < \frac{k-3}{k}$, we write

$$\begin{bmatrix} 2 - 2C - 6s_1^{(k-j)} < 0 \\ \cdots \\ 2 - 2C - 6s_j^{(k-j)} < 0 \\ 2 - 2C - 6s_{j+1}^{(j)} > 0 \\ \cdots \\ 2 - 2C - 6s_k^{(j)} > 0 \end{bmatrix}$$

and at $C > \frac{k-3}{k}$, we write similarly

$$\begin{bmatrix} 2 - 2C - 6s_1^{(k-j)} > 0 \\ \cdots \\ 2 - 2C - 6s_j^{(k-j)} > 0 \\ 2 - 2C - 6s_{j+1}^{(j)} < 0 \\ \cdots \\ 2 - 2C - 6s_k^{(j)} < 0 \end{bmatrix}.$$

Recall the saddle point criteria as $\exists_p \exists_q \; 2 - 2C - 6s_p^{sol} < 0 \wedge 2 - 2C - 6s_q^{sol} > 0$ and the maximum criteria as $2 - 2C - 6s^{sol} < \mathbf{0}$. By the above test criteria, for $C \neq \frac{k-3}{k}$, we have a saddle point for all $j \in [1, k-1]$ as well as a maximum for $j = k \wedge C < \frac{k-3}{k}$ and for $j = 0 \wedge C > \frac{k-3}{k}$ at

$$s^{max} := \begin{bmatrix} s_1^{(k-k)} & \cdots & s_k^{(k-k)} \end{bmatrix} = \begin{bmatrix} s_1^{(0)} & \cdots & s_k^{(0)} \end{bmatrix} = \begin{bmatrix} 1/k & \cdots & 1/k \end{bmatrix} \wedge C \neq \frac{k-3}{k}$$

since only at $j \in \{ 0, k \}$ do we have the case that either $s_p^{(j)}$ or $s_p^{(k-j)}$ is present in the solution $s^{sol}$.

At $C = \frac{k-3}{k}$, we have for any entry of $s^{sol}$ (here: $s_p^{(j')}$)

$$s_p^{(j', C = k - 3/k)} = \frac{-2(k-3)j'/k + 2j' - 3}{6j' - 3k} = \frac{6j'/k - 3}{6j' - 3k} = \frac{1}{k}.$$

Thus, although the second partial derivative test is indecisive since $2 - 2C - 6\,1/k = 0$, we have at $C = \frac{k-3}{k}$ always the same solution as in $s^{max}$. This renders $s^{max}$ for all $C$ as the maximal solution.

Next, we plug the solution $s^{max}$ into $f(s)$ and calculate the optimal front with the inequality constraint '$g_p(s) \leq 0$' of condition (2) and across all number of dimensions $k$ and range of the solution counter $j \in \{ 0, k \}$:

$$\max_{k,j} \left\{ f(s^{max}) \mid s_p^{(j)} \geq 0 \wedge s_p^{(k-j)} \geq 0 \wedge j \in \{ 0, k \} \right\}$$

$$= \max_k \left\{ \sum_{p=1}^{k} s_p^{(0)}(1 - s_p^{(0)})(C + s_p^{(0)}) \mid s_p^{(0)} \geq 0 \right\}$$

$$= \max_k \left\{ \sum_{p=1}^{k} \tfrac{1}{k}(1 - \tfrac{1}{k})(C + \tfrac{1}{k}) \mid \tfrac{1}{k} \geq 0 \right\}$$

$$= \max_k \left\{ C + \tfrac{1-C}{k} - \tfrac{1}{k^2} \right\}$$

$\Big($for $k = \frac{2}{1-C}$ the term $C + \frac{1-C}{k} - \frac{1}{k^2}$ is maximal for which we need the derivative to be zero: $\frac{d}{dk}(C + \frac{1-C}{k} - \frac{1}{k^2}) = \frac{C-1}{k^2} + \frac{2}{k^3} = 0\Big)$

$$= C + \tfrac{1}{2}(1 - C)^2 - \tfrac{1}{4}(1 - C)^2$$

$$= \tfrac{C^2}{4} + \tfrac{C}{2} + \tfrac{1}{4} = 0.25(C + 1)^2$$

Thus, we conclude that $f(s^{max})$ is equal to or below the convex hull $0.25(C + 1)^2$ for any solution counter $j$ and any number of classes $k$.

$\square$

Note: In this proof, we assumed $k \in \mathbb{R}_+$, however, we can restrict the number of classes $k$ even further: $k \in \mathbb{N}$ and $k \leq K$. Yet, this restriction does not have much impact on the bound on $f$ for a reasonable $C, K$: Now, we only have $K$ possible maxima ($s_p^{max} = \{ 1, 1/2, \ldots, 1/K \}$) where for a given $C$ only one of these maxima are dominant. This also means that our $0.25(C + 1)^2$-bound is a convex hull and only matches the maxima in a few selected points. However, already for little $K$ does the maximum come considerably close to the hull as shown in Fig. 9.

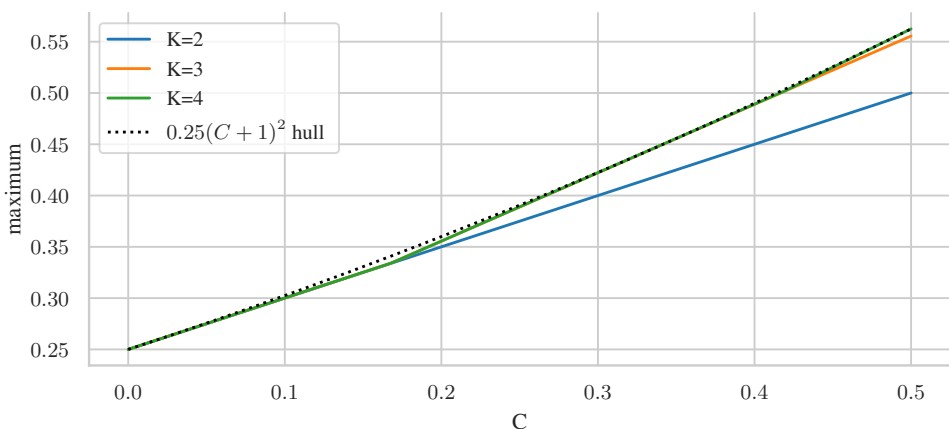

Figure 9: Precise maximum of $f(s^{max})$ per constant $C$ and restricted, discretized number of classes $k \leq K, k \in \mathbb{N}$ versus convex hull of the maximum of $f(s^{max})$ across all number of classes $k \in \mathbb{R}_+$.

