# OpenReview forum: "Distributed DPHelmet: Differentially Private Non-interactive Convex Blind Averaging"
_ICLR.cc/2024/Conference — Submitted to ICLR 2024_

### Official Review · Reviewer_jg5A · 2023-10-30

**Soundness:** 3 good
**Presentation:** 2 fair
**Contribution:** 2 fair
**Rating:** 5
**Confidence:** 3

**Summary:**

This paper proposes a novel federated learning algorithm based on differential privacy, secure multi-party computation and transfer learning. The goal is to minimize MPC Invocations. This work manages to use only 1 round of secure aggregation by taking averaging of local SVM models. Experiment results show that the proposed method achieves a better utility-privacy trade-off compared to DP-SGD.

**Strengths:**

The problem of privacy in federated learning is important. The proposed method is novel. Theoretical analysis is given for the proposed method.

**Weaknesses:**

There have been several existing works on the combination of differential-privacy and secure aggregation. Essentially you can turn local DP into a central DP via secure aggregation. In particular, [1] (which is already included in the submission) gives an implementation of distributed DP mechanism that has matching utility with central DP. In Table 1, you claim that DP-FL has noise scale of $O(1/m\sqrt{n})$, which could be too large.

If I understand correctly, 1 round of secure aggregation is enough because the average of local SVM models converges to global SVM optimal. This seems to limit the use case, because in practice we do not often use SVM in FL. Indeed we observe good performance in experiments, but it may also come from the use of pretrained feature extractor. More justification of the importance of SVM learning is highly appreciated.

[1] Kairouz, Peter, Ziyu Liu, and Thomas Steinke. "The distributed discrete gaussian mechanism for federated learning with secure aggregation." International Conference on Machine Learning. PMLR, 2021.

**Questions:**

typos:
- abstract, "...based on what we coin blind averaging", coin or call?
- Table 1, in the row of DP-FL, "− (O(M) rounds)", why do you have the "-"?

---

> ### Author Response · Authors · 2023-11-17
> **Rebuttal Response**
>
> We thank the reviewers for their insightful comments. Below, we will respond to some specific comments and refer to the shared comment above for a summary.
>
> ## Related work: Kairouz et al.
> > There have been several existing works on the combination of differential-privacy and secure aggregation. Essentially you can turn local DP into a central DP via secure aggregation. In particular, [1] (which is already included in the submission) gives an implementation of distributed DP mechanism that has matching utility with central DP. In Table 1, you claim that DP-FL has noise scale of O(1/(m sqrt(n))), which could be too large.
>
> We have updated the table in the updated submission, and have addressed the weaknesses of our original submission's Table 1 in a comment to reviewer 91JF.
>
> Kairouz et al. (2021) match the utility and privacy bound of centralized DP-SGD training (in our experiments: DP-FL with 1 user) but require too many MPC invocations as they noise and average the gradient in each iteration and not the parameter output once after training (as we do). It also remains open whether their MPC protocol is able to scale to millions of users as Bell et al. (2020) does. Thus, we excluded this bound in Table 1 due to this huge scalability issue with the number of iterations and users.
>
> ## Use of pre-training feature extractors
> > If I understand correctly, 1 round of secure aggregation is enough because the average of local SVM models converges to global SVM optimal. This seems to limit the use case, because in practice we do not often use SVM in FL. Indeed we observe good performance in experiments, but it may also come from the use of pretrained feature extractor. More justification of the importance of SVM learning is highly appreciated.
>
> Our results show strong utility and privacy bounds for scalable distributed SVM learning (with only 1 invocation of secure summation). Hence, we consider our results a valuable contribution to scenarios where SVM learning is sufficient. To the question, it is useful if foundation models [R1], hand-crafted features, or application-specific kernels are available.
>
> Additionally, we show strong distributed learning privacy results (privacy amplification via blind averaging due to an output sensitivity bound) for learning a Softmax-SLP, which is often used for fine-tuning the last layer of large neural networks.
>
> [R1] Rishi Bommasani et al. "On the opportunities and risks of foundation models." arXiv preprint (2021).
>
> ## Other comments
> > Table 1, in the row of DP-FL, "− (O(M) rounds)", why do you have the "-"?
>
> In Table 1 we have noted with "-" that DP-FL does not have MPC invocations but regular distributed training rounds. We updated Table 1 to clarify this point. In the response to reviewer 91JF, we have discussed the changes in Table 1 in more detail.

---

### Official Review · Reviewer_VsvY · 2023-10-30

**Soundness:** 2 fair
**Presentation:** 2 fair
**Contribution:** 3 good
**Rating:** 6
**Confidence:** 3

**Summary:**

/!\ the template ICLR is not used
This paper proposes to learn a private federated model with only one communication step. For doing so, it focuses on models where the final model can be computation by an averaging step (that could be done with secure aggregation), namely SVM and Softmax-activated single-layer perception. The paper assumes a fraction t of honest users, and only passive attackers. Differential Privacy is ensured by noise injection on the client side to the local model and by bounding the sensitivity for this model. Finally, the paper provides experiments on CIFAR10 and CIFAR100.

**Strengths:**

- The paper is clearly written and could easily be re-implemented and adapted to real use-case, the experiments are polished with all the parameters details and baselines included.
- The motivation of having a global model but with only one-communication round could make sense and having a precise analysis for SVM could be useful
- The solution is really privacy-oriented, because the approach uses the feature extractor to benefit from public knowledge rather than consuming privacy budget. Then, the strong convexity gives a very small sensitivity and finally reducing communication also reduce privacy loss. I really appreciate this design.

**Weaknesses:**

- The scope is narrowed to problems that can be learnt efficiently with SVM/SLP, and it scales poorly with the number of classes as seen on CIFAR100
- The privacy results doesn't seem a big contribution from the mathematical point of view, and the proofs are a bit messy (see below)
- The privacy results are only at the row level, and not at the user level. Overall, motivation and real-use cases for this privacy and communication setting could have been more developed
- No heterogeneity is tackled, despite the fact it is likely to be an issue is real-use cases

**Questions:**

- In my understanding, your setting would be quite nice for personalization: As every client learns the best model from itself, and as models can be averaged by designed, it could be worth to do a weighted averaging between global and local model, what do you think about it?
- Have you try to do experiments with some heterogeneity and other datasets? Even "fake" heterogeneity with just class unbalance would be a good complement
- I have quickly browsed the proofs, I saw that Lemma 24 is the usual technique of scaling by the sensitivity constant. The proof with developing the ratio seems unnecessary difficult and p27 in the calculation of the line starting by "due to the Cauchy-Schwarz inequality", you put square on the norms, but there are not. Maybe going through the appendix and simplify or highlights the key points could help
- I am not sure that the title is optimal to describe the paper, in particular it makes it sounds more theoretical.

---

> ### Author Response · Authors · 2023-11-17
> **Rebuttal Response (I)**
>
> We thank the reviewers for their insightful comments. Below, we will respond to some specific comments and refer to the shared comment above for a summary.
>
>
> ## Focus on SVM/Softmax-SLP leads to privacy amplification
> > The scope is narrowed to problems that can be learnt efficiently with SVM/SLP [...]
>
> This focus on SVM/Softmax-SLP allows us to prove an output sensitivity bound, which enables privacy amplification via blind averaging (Lemma 6) – a strong property for distributed learning. Such privacy amplification via blind averaging is not known for non-convex models. We even show an output sensitivity bound for fine-tuning large models with retraining their last layer with Softmax.
>
> Leveraging pre-training feature extractors has been proven promising for finding strong tradeoffs between expressivity, accuracy, and privacy guarantees ([R1], Tramèr and Boneh (2021), De et al. (2022)).
>
> [R1] Rishi Bommasani et al. "On the opportunities and risks of foundation models." arXiv preprint (2021).
>
>
> ## Softmax-SLP scales with number of classes
> > [Your algorithm] scales poorly with the number of classes as seen on CIFAR100.
>
>
> The problem you mention is valid for our Huber-loss SVM result. For handling multiple classes, we additionally provided privacy bounds for a Softmax-SLP (cf. Theorem 11; Corollary 12). Our Softmax-SLP performs competitively on a larger number of classes (cf. Figure 3): we reach 44.4\% on the 100 classes of CIFAR-100 ($\varepsilon=1.18$ and $100$ users) while comparable private methods like DP-FL only reach 27.3\%.
>
> The scenario that we consider is inherently difficult. We balance privacy, scalable distribution (of 100 or more users), and utility. In particular, we aim to minimize the number of MPC invocations to one. In this setting, we are not aware of any algorithm other than Softmax-SLP reaching a better accuracy.
>
>
>
>
> ## Non-trivial privacy results
> > The privacy results doesn't seem a big contribution from the mathematical point of view, and the proofs are a bit messy (see below)
>
> In terms of theoretical contribution, we provide the first output sensitivity result for Softmax-SLP (Theorem 11), by showing $L$-Lipschitzness (Theorem 27) and $\beta$-smoothness (Theorem 28) for Softmax-SLP.
>
> An output sensitivity bound is a sufficient condition for the privacy amplification via blind averaging (cf. Lemma 6). In other words, mechanisms with output sensitivity lead to privacy guarantees in a setting where the parties learn locally, but only add noise in the magnitude of centralized learning.
>
> ## Our person-level privacy
> > The privacy results are only at the row level, and not at the user level. Overall, motivation and real-use cases for this privacy and communication setting could have been more developed
>
> Our submission does present several flavors of person level privacy results, as we agree that person-level privacy is important. In use-cases where local aggregators collect data from people in their vicinity the row-based approach coincides with the person-level. In use-cases where all rows of a user belong to the same person, our submission does report user-level results: technically, we first showed guarantees for group-DP in Corollary 9 which then directly results in user level guarantees (cf. Corollary 10). We also evaluated this setting experimentally in Figure 5 (Appendix A) which demonstrates the applicability of our method to user-level privacy as long as there are enough users. In this evaluation, each user holds 50 data points, but the number (and exact training procedure) can differ from one user to another.
>
> ## DPHelmet's heterogeneity
> > No heterogeneity is tackled, despite the fact it is likely to be an issue is real-use cases. Have you tried to do experiments with some heterogeneity and other datasets? Even 'fake' heterogeneity with just class unbalance would be a good complement
>
> We tackled a strongly-biased non-iid scenario in Table 2 which represents one of the worst-case distributions of data points among users. Here each user has only exclusive access to one class.
>
> We would like to emphasize that privacy-preserving learners have been shown to lead to discrimination against minority classes [R2].
>
> [R2] Eugene Bagdasaryan, Omid Poursaeed, and Vitaly Shmatikov. "Differential privacy has disparate impact on model accuracy." NeurIPS, 32, 2019.
>
> ## Usecase: Personalization
> > In my understanding, your setting would be quite nice for personalization: As every client learns the best model from itself, and as models can be averaged by designed, it could be worth to do a weighted averaging between global and local model, what do you think about it?
>
> This is an interesting idea, thank you a lot! We agree that DPHelmet might perform well in such cases. We will add a suggestion for future work to investigate this in our discussion section.

---

> ### Author Response · Authors · 2023-11-17
> **Rebuttal Response (II)**
>
> ## Our Proofs
> > I have quickly browsed the proofs, I saw that Lemma 24 is the usual technique of scaling by the sensitivity constant. The proof with developing the ratio seems unnecessary difficult and p27 in the calculation of the line starting by "due to the Cauchy-Schwarz inequality", you put square on the norms, but there are not. Maybe going through the appendix and simplify or highlights the key points could help.
>
> We use Lemma 24 to prove the bound in our group privacy variant (cf. Corollary 9), which we use to prove that user-level privacy holds with strong utility-privacy tradeoffs. We agree that the argumentation can be simplified by analyzing the sensitivity in the group privacy setting.
>
> At p27 we take the square root on the norm to allow us to put squares on the norms.
>
> ## ICLR template
> > /!\ the template ICLR is not used.
>
> Thanks for pointing that out! This was our mistake, we now added the ICLR header to the paper.

---

> > ### Comment · Reviewer_VsvY · 2023-11-20
> > **thank for rebuttal**
> >
> > I thank the authors for their detailed rebuttal. As they answered my concerns, even if some other interesting concerns has been raised by other reviewers, I raise my score.

---

### Official Review · Reviewer_91JF · 2023-11-01

**Soundness:** 2 fair
**Presentation:** 1 poor
**Contribution:** 1 poor
**Rating:** 3
**Confidence:** 4

**Summary:**

This paper studies the problem of differentially private distributed optimization. More specifically, the authors consider the setting where only one round of communication is allowed during the optimization. The authors consider the private variant of two specific methods, i.e., SVM and Softmax-SLP in this setting.

**Strengths:**

The strengths of the current paper:
1. The problem considered in this paper, i.e., secure aggregation+differential privacy, is interesting and promising.

**Weaknesses:**

The weaknesses of the current paper:
1. The presentation of the current paper is unclear, and it is very hard to follow the results and discussions.
2. It is unclear whether the utilities for different methods in Table 1 are correct.
3. The computation and memory cost of the secure summation is unclear.

**Questions:**

I have the following addition concerns besides the weaknesses:
1. In Table 1, where are the references for the DP-FL, Centralized training, and the utility for your proposed method?
2. I find it is very hard to follow the main results in the current paper. For example, in Corollary 4, why do you have the constraint on $\epsilon$ and what is the meaning of adding random noise to a set of outputs and why do you need to define $I_d$?
3. I don't understand the claim about tight composition results under Corollary 4.
4. How will the number of local updates $M$ affect your privacy and utility guarantees?
5. How will the rescaling step in your algorithms (e.g., projected SGD) affect your utility guarantees?
6. What is SimCLR, and how will it affect your results?
7. What is the definition of the honest user? In addition, why the noise magnitude will be reduced by a factor of $t$ when you have $t$ fraction of honest users?
8. How do you implement secure aggregation?
9. I don't understand $\nu$ in Theorem 8.
10. I don't understand Theorem 14, what do you mean by the results belongs to $O(1/M)$?

---

> ### Author Response · Authors · 2023-11-17
> **Rebuttal Response (I)**
>
> We thank the reviewers for their insightful comments. Below, we will respond to some specific comments and refer to the shared comment above for a summary.
>
> ## More helpful utility bounds in Table 1
> > It is unclear whether the utilities for different methods in Table 1 are correct.
>
> We agree that our statements in Table 1 on the utility bounds were not helpful. We have replaced those with what we deem more relevant and to the point: whether or not blind averaging works, i.e. whether, if no noise is added, the approach converges to the model we would train centrally.
>
> DP federated learning (DP-FL) and Jayaraman et al.'s gradient perturbation just train a global model which by definition is the same as the centralized one. For our SVM result, we have shown this convergence formally in Theorem 14. For our Softmax-SLP result, we do not have a formal proof, but have indicated it experimentally and so have Jayaraman et al. for their output perturbation result. The convergence rate of Jayaraman et al. (2018, Theorem 3.2) depends on the number of data points whereas we truly converge with the number of iterations (cf. Theorem 14).
>
> We also divided Table 1 by the algorithms, i.e. SVM and Softmax-SLP, which have different use-cases and face different challenges. For more details on these differences see our answer for reviewer VsvY ("Softmax-SLP scales with number of classes" and "Non-trivial privacy results").
>
>
>
> ## Cost of secure summation
> > The computation and memory cost of the secure summation is unclear.
>
> The computation and memory costs of secure summation depend on the particular secure summation variant. If you use Bell et al. (2020) we provided the computation costs in the extended preliminaries in Appendix C.2 and for the example of DPHelmet in Section 6, "Computation costs". Most notably, in DPHelmet's SVM-SGD ($1000$ users) we only transmit data of $l \approx 100{,}000$ floating points.
>
> We have not measured the memory costs of Bell et al.'s protocol.
>
>
> ## References for Table 1
> > In Table 1, where are the references for the DP-FL, Centralized training, and the utility for your proposed method?
>
> Centralized training refers to Algorithm 1 (SVM) and 3 (Softmax-SLP) respectively. As discussed in our introduction (cf. Section 1) and extended related work in Appendix D.1, DP-FL refers to a combination of federated learning (McMahan et al., 2017) with DP-SGD (Abadi et al., 2016). For the utility of our proposed method, see above "Utility bounds in Table 1".
>
> ## Corollary 4
> > I find it is very hard to follow the main results in the current paper. For example, in Corollary 4, why do you have the constraint on eps and what is the meaning of adding random noise to a set of outputs and why do you need to define $I_d$?
>
> Corollary 4 directly follows from Lemma 3 where we rearranged the noise scale restriction to a restriction on the epsilon. We add $(p+1)\times |K|$-dimensional noise to a $|K|$-sized set of $(p+1)$-dimensional outputs. Formally, we add for each element in the $|K|$-sized set $(p+1)$-dimensional noise. We defined $I_d$ to prevent any confusion of our theorems.
>
> ## Tight composition
> > I don't understand the claim about tight composition results under Corollary 4.
>
> Corollary 4 is not tight, i.e. we can find for a given delta an epsilon that is smaller than proven in Corollary 4 although $(\varepsilon,\delta)$-DP (Definition 16) is still guaranteed. This is a particular result since we use additive Gaussian noise as a privacy mechanism. Additionally, we train $|K|$ SVMs which represents a $|K|$-fold sequential composition of a Gaussian mechanism that can be shown to scale with $\mathcal{O}(\sqrt{|K|})$ which is significantly better than the naïve sequential composition bound with a scale multiplier of $|K|$ in Corollary 4. For more details, we refer to the work of Sommer et al. (2019), and Balle et al. (2020a).
>
> ## Effect of local updates on privacy/utility
> > How will the number of local updates M affect your privacy and utility guarantees?
>
> The privacy guarantees are independent of the number of local updates $M$ since we consider convex, Lipschitz, and $\beta$-smooth optimization problems. In Lemma 2, we recall Wu et al. (2018, Lemma 8) that show for a strongly convex optimization objective, e.g. SVM, LR, or Softmax-SLP, with a linear decreasing learning rate, that the noise scale is independent of the number of iterations M.
>
> Concerning utility, our convergence result of SVM-training in Theorem 14 shows that the average of locally trained SVM converges with rate $1/M$ to the best model for the combined local datasets $\mho$ where $M$ are the number of local iterations.

---

> ### Author Response · Authors · 2023-11-17
> **Rebuttal Response (II)**
>
> ## Projected SGD
> > How will the rescaling step in your algorithms (e.g., projected SGD) affect your utility guarantees?
>
> Projected SGD is a requirement for the convergence rate of $1/M$ (cf. Proof of Theorem 14 and in particular the result of Locoste-Julien et al., 2012) as well as for a useful bound on differential privacy (cf. Wu et al., 2017). For privacy, we need a bounded hypothesis space which is also reflected by the noise scale which depends on the radius of the hypothesis space $R$. With projected SGD we can enforce such a bounded hypothesis space.
>
>
> ## SimCLR
> > What is SimCLR, and how will it affect your results?
>
> The effect of SimCLR is illustrated in Figure 6 in Appendix C.3 and described there as well. In essence, it reduces the dimensionality of the image data via a deep neural network that has been trained on ImageNet. The prediction of SimCLR gives us a $6{,}144$d embeddings vector on which we perform SVM, Softmax-SLP, and DP-FL. This technique goes in hand with the privacy literature (Tramèr and Boneh (2021), De et al. (2022)) to boost the privacy-utility tradeoff.
>
> ## Honest users
> > What is the definition of the honest user? In addition, why the noise magnitude will be reduced by a factor of t when you have t fraction of honest users?
>
> An honest user does not adversely collaborate with other users which means in our case that it adds the required Gaussian noise appropriately. Thus we reduce the noise by the fraction of honest users since we only assume that a share of $t$ of the added noise is actually added.
>
> ## Secure Aggregation Implementation
> > How do you implement secure aggregation?
>
> The implementation of a concrete secure aggregation protocol is not in the scope of this work.
> Also note, that our work does not require a specific secure summation technique and is agnostic to different secure summation implementations.
>
> ## $\nu$ in Theorem 8
> > I don't understand nu in Theorem 8.
>
> Since we use MPC, we have to use the computational differential privacy notion (cf. Definition 17). $\nu$ is the additional contribution to $\delta$ defined in the full version of Theorem 8 in Appendix J. Computational differential privacy has a function $\nu_0$ negligible in the security parameter $\eta$ used in secure summation and our $\nu$ directly depends on this function $(1+e^\varepsilon)\cdot \nu_0$, where $\varepsilon \ll 10$ is a small number. Thus, it remains negligible.
>
> ## Theorem 14: O(1/M)
> > I don't understand Theorem 14, what do you mean by the results belongs to O(1/M)?
>
> The convergence rate is in proportional to $1/M$, which we formalized by using the asymptotic $\mathcal{O}$-notation, where $M$ is the number of iterations in the local training.

---

### Official Review · Reviewer_ooDC · 2023-11-01

**Soundness:** 3 good
**Presentation:** 2 fair
**Contribution:** 3 good
**Rating:** 5
**Confidence:** 4

**Summary:**

This paper aims at exploiting the property that in part of hyperparameter space the average of SVM models learnt on partial datasets is the same as the model learnt on the full dataset, and hence in a federated setting a single secure aggregation operation is sufficient to combine local models into a global model.

**Strengths:**

In scenarios (data sets, problems) where this property holds, reducing the number of MPC rounds is indeed an important gain.

The paper is understandable, even though the presentation is not perfect.

**Weaknesses:**

The key limitation (also explicitly mentioned in Section 7) is that "there exists a regularization parameter \Lambda" means that potentially (for some scenarios, data sets, problems) the only values of \Lambda for which the proposed technique works are unsuitable values of \Lambda, i.e., values which don't lead to a satisfactory model.  While this limitation is recognized, the paper does little effort to investigate when \Lambda values for which the proposed technique works also are satisfactory and lead to good models.  While the paper shows a small empirical evaluation, it isn't fully clear how widely applicable the proposed methods can be.


There are quite a number of points where the text is insufficiently precise.  E.g., only already in Section 2:
* Contributions: (1) Output sensitivity suffices for strong privacy results in blind averaging. : O((\cup_i D_i)^{-1}) : you can't invert a set, only the size of a set.  Why not say O((\sum_i |D_i|)^{-1}) ?
* "the size of each communication round" : Please define the "size of a round".  There are probably less ambiguous formulations such as "the communication cost of a round", "the computation cost of a round", "the number of messages in every round", ....
* "It does need a communication round per training iteration M." -> M is undefined and looks here like a variable representing a training round.  From the use of M much later in the text, I guess you mean "Let M be the number of training iterations.  It (the algorithm?) needs only one communication round per training iteration."
* Table 1: where log(M) is used, looking in the cited paper suggests you probably mean the logarithm of the number of users rather than the number of training iterations, but even then this seems to represent the number of rounds per training iteration rather than the global number of MPC invocations.
* Algorithm 1: parameter h is taken as input but never occurs (explicitly) in the code of the algorithm.  Probably it is some implicit parameter to the l_huber function.
* Algorithm 1: in every iteration, f_m^{(k)} is computed, but one would expect that f_m^{(k)} depends on f_{m-1}^{(k)}, i.e., the result of the previous iteration.  This can't be seen in the code.
* Just after Algorithm 1: How do you get to the specific number of "1920 rounds?"

**Questions:**

* Just after Algorithm 1: How do you get to the specific number of "1920 rounds?"  (more generally, I understand little of the provided argument here as the cited papers not always allow for finding easily the claim for which they are cited).

* What evidence is there is the extent to which the proposed method is applicable to more than just a few simpler datasets satisfying some desirable properties (being balanced, having little noise, being separable, ...) ?

**Details Of Ethics Concerns:**

--

---

> ### Author Response · Authors · 2023-11-17
> **Rebuttal Response**
>
> We thank the reviewers for their insightful comments. Below, we will respond to some specific comments and refer to the shared comment above for a summary.
>
> ## Evidence for Convergence on smaller $\Lambda$s
> > The key limitation (also explicitly mentioned in Section 7) is that "there exists a regularization parameter $\Lambda$" means that potentially (for some scenarios, data sets, problems) the only values of $\Lambda$ for which the proposed technique works are unsuitable values of $\Lambda$, i.e., values which don't lead to a satisfactory model. While this limitation is recognized, the paper does little effort to investigate when $\Lambda$ values for which the proposed technique works also are satisfactory and lead to good models. While the paper shows a small empirical evaluation, it isn't fully clear how widely applicable the proposed methods can be.
>
> The need for large $\Lambda$s for output sensitivity of convex optimization (including SVMs) has already been identified by Chaudhuri et al. (2011) since the output sensitivity and thus the noise scales with $1/\Lambda$. Also, for the experiments, choosing larger $\Lambda$s is useful as it allows a smaller sensitivity thus less noise. Output sensitivity is important for privacy amplification via blind averaging (Lemma 6).
>
> From the theoretical perspective, our results provide a tool for analyzing utility bounds from the perspective of the dual problem.
>
> ## Generalizability of DPHelmet
> > What evidence is there is the extent to which the proposed method is applicable to more than just a few simpler datasets satisfying some desirable properties (being balanced, having little noise, being separable, ...) ?
>
> For SVMs, we show in Theorem 14 a general convergence statement for SVM for a large enough $\Lambda$. Our non-iid intuition in Figure 2 also demonstrates this on a highly unbalanced, non-separable dataset.
>
> Currently, all our experiments speak in our favor as CIFAR-10 and CIFAR-100 datasets are highly unbalanced on a per-SVM basis (for SVMs, we use the one-vs-rest approach, where we train n\_classes SVMs and weigh each class against all others). For SVMs and Softmax-SLP, we analyzed in Table 2 how our method performs in a setting where the local training data sets are unbalanced and non-iid variants of the CIFAR-10 and CIFAR-100 datasets.
>
> We would like to emphasize that privacy-preserving learners have been shown to lead to discrimination against minority classes [R2].
>
> Good separability can often be offered by foundation models like SimCLR (Chen et al. 2020b) where the authors suggest fine-tuning a linear layer. These foundation models have gained increasing popularity and relevance in practice [R1].
>
> We are not sure what you meant by "having little noise". Requiring little DP noise is not an issue in our experiments as we have strong results for $\varepsilon=0.1$ (cf. Figure 3) and even $\varepsilon=2 \cdot 10^{-5}$ (cf. Figure 5). This is due to our sensitivity bound which scales by $\frac{1}{nm}$.
>
> [R1] Rishi Bommasani et al. "On the opportunities and risks of foundation models." arXiv preprint (2021).
> [R2] Eugene Bagdasaryan, Omid Poursaeed, and Vitaly Shmatikov. "Differential privacy has disparate impact on model accuracy." NeurIPS, 32, 2019.
>
>
> ## Impreciseness in Section 2
> We clarified all your points throughout our paper in an updated version.
> - We agree, $O((\sum_i |D_i|)^{-1})$ is better.
> - With "size of each communication round" we meant "communication cost of each round".
> - Yes, M refers to the number of iterations. This means that DP-FL needs one communication round per training iteration.
> - You are correct: the number of MPC invocations in Jayaraman et al. (2018, Theorem 3.6) scales with $\mathcal{O}(\log(nm))$ to reach their utility bound. Note that in our scheme we can train until convergence without sacrificing utility by stopping early. They have to stop early since their noise scales with the square root of the number of iterations (Theorem 3.4 in Jayaraman et al. (2018)). In their Section 3.2 above Theorem 3.4, they stated that they perform one MPC invocation per iteration.
> - The parameter h in Algorithm 1 scales the smoothness of the Huber loss (cf. Definition 21).
> - We are now more precise in Algorithm 1: we added $f^{(k)}_{m-1}$ as an additional input to the function "SGD".
> - $1920$ is the number of iterations we used for DP-FL and is loosely adapted to the ones reported by Tramèr and Boneh (2021); it consists of $40$ epochs and $48$ iterations per epoch ($40\cdot48=1920$).

---

> > ### Comment · Reviewer_ooDC · 2023-11-18
> >
> > Thanks for your clarifications.
> >
> > I feel this paper is interesting.  Still, the results shown require that Lambda is sufficiently large (which apparently you already expected given earlier work).   A major concern is that a user who wants to use this in practice does not seem to have an easy criterion to decide whether for his problem the optimal result can be achieved with sufficiently high Lambda, in which case he can use your theory, or whether the optimal result will have a lower Lambda for which he may today not yet have a sufficiently satisfactory solution.  Of course, he could try various values of Lambda and see what are the results, but that then requires he makes the investment of running the method already before he knows whether he will be able to benefit from your results.

---

> > > ### Author Response · Authors · 2023-11-20
> > >
> > > > Still, the results shown require that Lambda is sufficiently large (which apparently you already expected given earlier work). A major concern is that a user who wants to use this in practice does not seem to have an easy criterion to decide whether for his problem the optimal result can be achieved with sufficiently high Lambda, in which case he can use your theory, or whether the optimal result will have a lower Lambda for which he may today not yet have a sufficiently satisfactory solution. Of course, he could try various values of Lambda and see what are the results, but that then requires he makes the investment of running the method already before he knows whether he will be able to benefit from your results.
> > >
> > > The problem of finding the best hyperparameters is indeed an open problem for many differentially private machine learning algorithms. Many popular DP ML mechanisms, such as DP-SGD (e.g., implemented in Opacus or Tensorflow Privacy) require hyperparameter tuning for optimal performance. Via rejection sampling, it is possible to limit this leakage of a hyperparameter search to roughly a factor of 2-3 in terms of epsilon [R3]. Finding a fully hyperparameter-free variant of our method is an interesting direction for future work but out of scope of our current work.
> > >
> > > As a first step, our submission presents a version of SVM-SGD (in Appendix B) that does not require more than two hyperparameters since of the two relevant hyperparameters one ($R$) does not have a large influence on the sensitivity and the other hyperparameter ($\Lambda$) is approximately linearly linked to $\varepsilon$.
> > >
> > > In settings where the datasets are balanced per-SVM (i.e., per user) and regularization is acceptable, our result for blindly averaged hinge-loss SVMs provides even more information (cf. convergence theorem, Theorem 14). In such settings, our convergence theorem states that we are in the same setting as for global SVMs. In particular, if you set $\Lambda = 20{,}000{,}000$ then you will receive the same hyperplane (scaled by $1/1{,}000{,}000$) as for $\Lambda=20$. Thus a practitioner can -- by our theoretical findings -- set $\Lambda$ to e.g. $20{,}000{,}000$ and receive convergence (the input space is because of privacy bounded thus there will always be a large enough $\Lambda$).
> > > The reason for the hyperplane equivalence is that at $\Lambda=20$ all data points are support vectors (cf. Figure 2). By the representer theorem, the hyperplane $f$ can be characterized as $f=\sum_{(x_i,y_i)\in D} \alpha_i x_i y_i$ (cf. Theorem 20 where we extracted $y_i$ from the $\alpha_i$ for readability), and by Ma & Ng (2020, Equation 28-30) all $\alpha_i$ are equal to $\Lambda$. Thus we reach the same hyperplane scaled by the ratio of the old $\Lambda$ and the new $\Lambda$.
> > >
> > > [R3] Jingcheng Liu and Kunal Talwar. Private Selection of Private Candidates. In STOC, 2019.

---

### Author Response · Authors · 2023-11-17
**Rebuttal Comment**

We thank all reviewers for their insightful comments. We updated our paper to address many of your presentation comments and to clarify our argumentation.

We answer each reviewer individually. As a summary, the individual answers cover and try to clarify potential misunderstandings from the reviewer comments on the following topics:

- Evidence for Convergence on smaller $\Lambda$s (ooDC)
- Generalizability of DPHelmet (ooDC)
- More Helpful utility bounds in Table 1 (91JF)
- Focus on SVM/Softmax-SLP leads to privacy amplification (VsvY)
- Softmax-SLP scales with number of classes (VsvY)
- Non-trivial privacy results (VsvY)
- Our person-level privacy (VsvY)
- DPHelmet's heterogeneity (VsvY)
- Use of pre-training feature extractors (jg5A)

---

### Meta-Review · Area_Chair_xsfd · 2023-12-07

**Metareview:**

Unfortunately, the paper did not get enough support from the reviewers. There were repeated concerns about the presentation (including the presentation of the bounds). I too took a closer look a the paper, and it was hard to tease apart the main contribution. In particular, the paper does not compare to the results achieved by https://www.ieee-security.org/TC/SP2017/papers/373.pdf in a similar space of problems. My concern is that the results that the paper proposes, can be easily achieved by standard random projection, and computing summary statistics in those dimensions (since the loss they consider is a generalized linear model (GLM)). To demonstrate non-triviality of the approach the paper should provide clear justifications why such simple methods would not suffice and cannot achieve the bounds the paper proposes.

**Justification For Why Not Higher Score:**

The main contributions of the paper are hard to parse, and the reviewers were not supportive of the paper.

**Justification For Why Not Lower Score:**

NA

---

### Decision · Program_Chairs · 2024-01-16

Reject